# Controlling the motion of multiple objects on a Chladni plate

Quan Zhou[1,*], Veikko Sariola[1,2,*], Kourosh Latifi[1,*] & Ville Liimatainen[1]

The origin of the idea of moving objects by acoustic vibration can be traced back to 1787, when Ernst Chladni reported the first detailed studies on the aggregation of sand onto nodal lines of a vibrating plate. Since then and to this date, the prevailing view has been that the particle motion out of nodal lines is random, implying uncontrollability. But how random really is the out-of-nodal-lines motion on a Chladni plate? Here we show that the motion is sufficiently regular to be statistically modelled, predicted and controlled. By playing carefully selected musical notes, we can control the position of multiple objects simultaneously and independently using a single acoustic actuator. Our method allows independent trajectory following, pattern transformation and sorting of multiple miniature objects in a wide range of materials, including electronic components, water droplets loaded on solid carriers, plant seeds, candy balls and metal parts.

[1] Department of Electrical Engineering and Automation, School of Electrical Engineering, Aalto University, Maarintie 8, Espoo 02150, Finland. [2] Department of Automation Science and Engineering, Tampere University of Technology, Korkeakoulunkatu 3, Tampere 33720, Finland. * These authors contributed equally to this work. Correspondence and requests for materials should be addressed to Q.Z. (email: quan.zhou@aalto.fi).

Moving objects by acoustic vibration on solid surfaces or in fluids has a myriad of applications in manipulating cells, droplets and particles[1–9]. Early studies of moving particles and forming patterns on vibrating plates were performed already hundreds of years ago by Leonardo da Vinci, Galileo Galilei, Robert Hooke and Ernst Chladni[10–12]. In his famous experiment in 1787, Chladni drew a bow over a piece of centrally fixed metal plate covered with sand, and the vibration of the plate caused the sand to move and accumulate around the nodal lines where the surface remained still, forming Chladni figures[12]. The experiments by Chladni are a corner stone of modern acoustics. However, the motion of particles before they settle to the nodal lines is still not very well understood, and only hypothetical models have been put forward[13]. The prevailing view has been that the particle motion out of nodal lines is random[13,14], implying uncontrollability[15]. Consequently, modern acoustic manipulation methods have focused on controlling the position and shape of nodes[1,2,6,8,16–20]. State-of-the-art acoustic manipulation systems are, however, either rather limited in dexterity[21] or relatively complex in terms of hardware. Extensive work can be found on acoustic linear motors and actuators with one degree of freedom[22,23]. Four actuators have generated surface acoustic traps to control the two-dimensional motion of single particles[6]. Four actuators have also created a local velocity field to control the motion of multiple individual objects one by one[24,25], and six actuators can create programmable velocity fields to move multiple objects in a dependent manner[26]. In a more dexterous system, 64 ultrasonic channels have been used for simultaneous independent motion of up to three particles[2].

In the present study, we introduce a method for simultaneously and independently controlling the motion of multiple objects on a Chladni plate with only a single acoustic actuator (Fig. 1a and Supplementary Movie 1). A statistical model is developed to predict the object displacement for different notes in western musical scale. We control the position of multiple objects simultaneously and independently by using the model to select a note to play at each time step. This new understanding of object motion on Chladni plate greatly advances acoustic manipulation: we are able to control more degrees of freedom using just a single actuator compared with the recent demonstration using 64 actuators[2]. We demonstrate independent trajectory following, pattern transformation and sorting of multiple objects. A wide range of miniature objects is manipulated, including electronic components, water droplets loaded on solid carriers, plant seeds, candy balls and metal parts.

## Results

**Displacement field modelling and motion control.** The schematic of our experimental platform is shown in Fig. 1a and Supplementary Movie 1. The apparatus consists of a thin silicon plate $(50 \times 50 \times 0.525 \, mm)$, actuated by a centrally mounted piezoelectric actuator. The objects to be controlled are placed on top of the plate, and imaged by a camera. A computer detects the positions of the objects using machine vision. On the basis of the detected positions, the computer decides which direction it should move the objects next, and then finds the frequency of a note that is expected to most likely achieve the desired motion. The note is then played on the piezoelectric actuator, and the next control cycle is started based on the new positions of the objects. Control cycles are repeated until the objects have reached their desired target locations. Throughout this paper, the played notes were

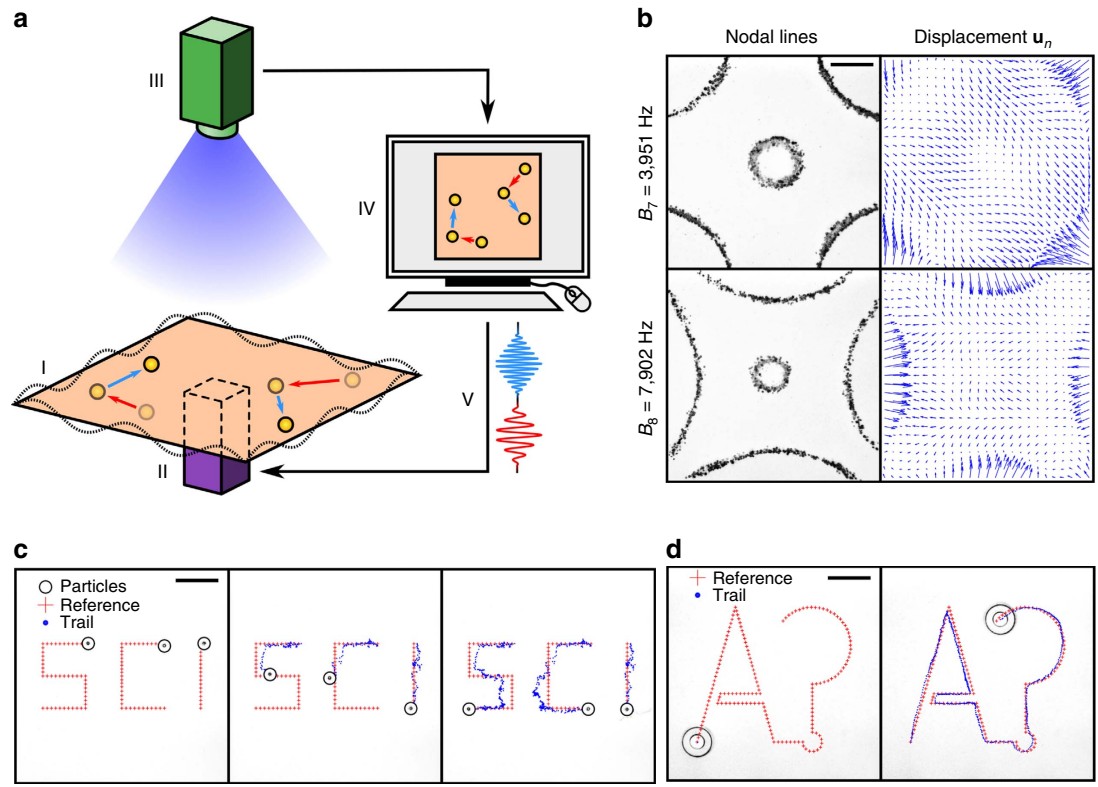

**Figure 1 | Concept of the manipulation method.** (**a**) Schematic of the experimental set-up: the set-up consists of a vibrating plate (I) with the manipulated objects on top, mounted on a piezoelectric actuator (II). The plate is imaged by a camera (III), and the locations of the objects are detected by a computer (IV). The computer chooses next note (V) to be played, so the objects move approximately towards their desired directions. (**b**) Chladni figures and modelled displacement fields for two different frequencies. (**c**) Simultaneous manipulation of three 600 μm solder balls. (**d**) Manipulation of a 7 mm washer. Scale bars, 1 cm.

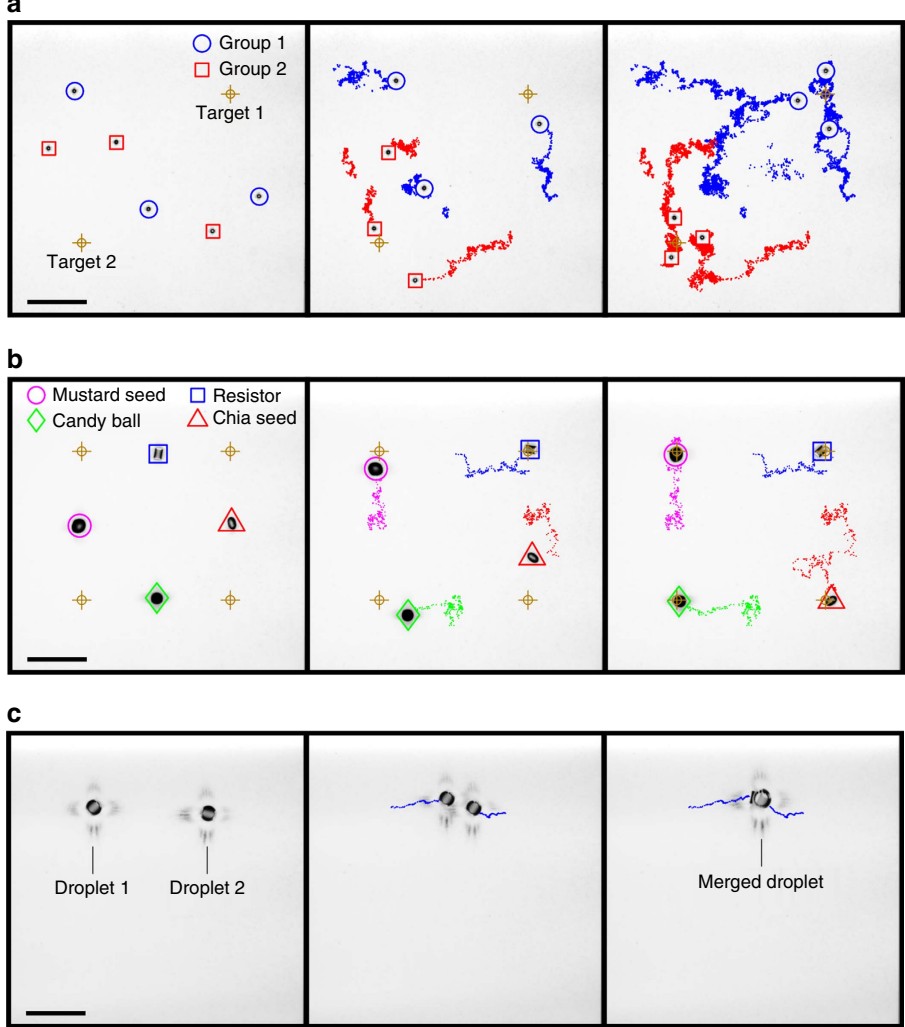

**Figure 2 | Multi-particle manipulation demonstrations.** (**a**) Particle sorting. Six 600 μm solder balls are assigned to two groups, three particles each. The device separates the particles into the assigned groups. (**b**) Particle pattern transformation. A diamond pattern of four objects—a SMT resistor, a mustard seed, a chia seed and a candy ball—is transformed into a square pattern. (**c**) Droplet merging. Two 8 μl water droplets on carriers are brought together and merged into a single droplet. Scale bars, 1 cm.

sinusoids with a triangular envelope and a duration of 0.5 s, unless otherwise stated (see Methods for details of the signal shape).

To predict the displacement of the objects, we track the positions of particles (600 μm solder balls) before and after playing a note, for all $N = 59$ different notes. The number of particles on the plate is 131 on average, with each note being played 50 times, resulting in ~6,600 data points per note and 390,000 data points in total. For each note, we fit a model to the data:

$$\Delta \mathbf{p} = \mathbf{u}_n(\mathbf{p}) + \mathbf{e} \tag{1}$$

where $\mathbf{p} \in \mathbf{R}^2$ is the position of a particle before playing a note, $\Delta \mathbf{p} \in \mathbf{R}^2$ is the displacement of the particle after playing note $n$, $n \in 1..N$, $\mathbf{u}_n$ is a two-dimensional displacement model fitted for note $n$ and $\mathbf{e}$ is the residual. We use locally weighted scatterplot smoothing (LOESS) regression[27] for $\mathbf{u}_n$. Details of the modelling are explained in Methods. Figure 1b shows $\mathbf{u}_n$ for two of the frequencies (see Supplementary Fig. 1 for all notes). The nodal lines correspond well to the Chladni figures created with continuous sinusoids (Fig. 1b), indicating that the finite duration and enveloping of the notes do not significantly alter the figures. In summary, we model $\mathbf{u}_n$ that captures not only the locations of the nodal lines but also the motion of the particles when they are far from the nodal lines.

We control the motion of the objects on the plate by repeatedly measuring the position of the objects and use the model to choose a note that moves the objects towards their desired directions. In each time step, the computer finds a linear combination of all the modelled displacement fields (Supplementary Fig. 1) such that all the weights are positive[28] and the net motion for all the objects at their current position is guided towards the desired direction. The new weight of each note is then added to the previously accumulated weight of that note. Then the note with the highest accumulated weight is played and its accumulated weight is reset to zero. The procedure is repeated in the following time steps until the targets are reached. As a result, notes that have generally large weights will be played more often, but occasionally notes with small but non-zero weights will also be played. The details of the control method are explained thoroughly in Methods, as well as in Supplementary Movie 2.

Our method can be applied to direct objects along trajectories. We split the trajectories into a set of waypoints and set the desired direction of each object towards its current waypoint. When the object is close enough to the waypoint, next waypoint along the

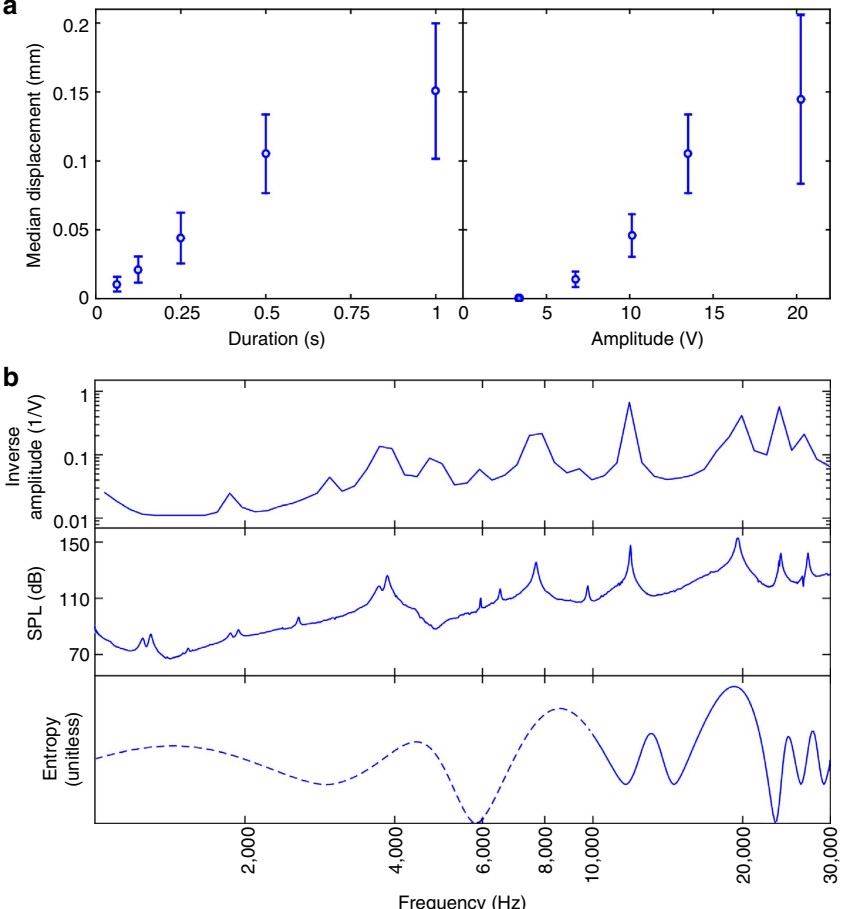

**Figure 3 | Dependence of the particle motion on actuation signal parameters.** (**a**) Median displacement of a 600 µm solder ball when varying duration and amplitude at 3,990 Hz. The error bars show median standard error, where the median is taken over the whole plate. (**b**) Inverse nominal amplitude, sound pressure level (SPL) and entropy as a function of frequency. The theoretical entropy curve is not expected to be valid at low frequencies (dashed line).

trajectory is chosen. We have successfully performed trajectory following with one and three solder balls simultaneously (Fig. 1c, Supplementary Fig. 2 and Supplementary Movie 3). The method is also not limited to only manipulating sub-mm particles. The largest object that we have manipulated was a 7 mm-diameter washer (Fig. 1d and Supplementary Movie 4), corresponding to 14% of the lateral dimensions of the plate. It is noteworthy that the manipulation was successful even though the model was developed using solder ball particles, thus demonstrating that the model is not highly specific to the shape and dimensions of the objects.

We have demonstrated our method in several practical applications. Figure 2a shows the sorting of six particles into two distinct groups (Supplementary Movie 5). Three of the particles have goal points towards the upper right corner of the plate, while three of the particles have goal points towards the lower left corner of the plate. At the end of the experiment, the particles are well divided into two distinct groups. We have also demonstrated pattern transformation, where we transform a diamond shape, defined by the four corner points, into a square shape (Fig. 2b and Supplementary Movie 6). Four different types of miniature objects are used: a mustard seed; a chia seed; a candy ball; and a surface mount technology (SMT) resistor. Besides solid objects, we have also transported water droplets on solid carriers (Fig. 2c and Supplementary Movie 7). Two carriers each loaded with an 8 µl water droplet were transported over a distance and finally merged into a single droplet. In addition, we have demonstrated aligning six radio frequency identification chips in

a line from initially scattered formation (Supplementary Fig. 3) and simultaneous trajectory following of two SMT resistors along complicated paths (Supplementary Movie 8).

**Effects of driving signal parameters on the displacements**. The displacement of the objects was found to vary with the duration, amplitude and frequency of the played note. Figure 3a shows the displacement for one frequency as a function of duration and amplitude. The observed relationship with duration is roughly linear, while the relationship with amplitude is close to linear but having a dead zone. At small amplitudes, we have observed a dead zone where no displacement was observed; we attribute this to static friction. Taken together, the amplitude and duration results show that either the amplitude or the duration of the notes can be adjusted to control the displacement caused by each note during the manipulation.

The frequency dependence is far more complicated, and is related to the resonances of the plate: at resonance, much smaller actuation amplitudes are needed to move the particles. For controlled manipulation, we want to keep the particle displacement relatively constant for all notes. To do this, we keep the duration of the notes constant (0.5 s), and algorithmically adjust the amplitude of each note such that the 75% quantile absolute displacement, taken over ∼100 particles distributed on the plate, is close to 175 µm for all notes. We call this adjusted amplitude the nominal amplitude for note $n$ (see section Nominal amplitude in Methods).

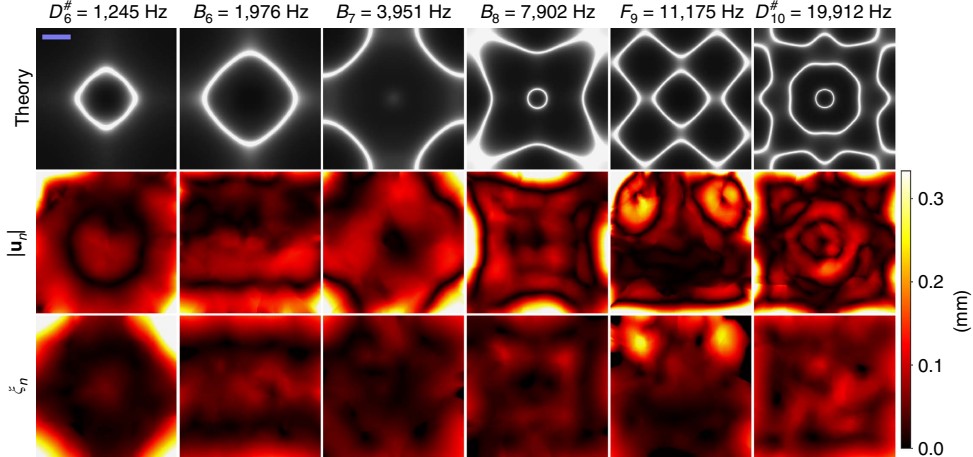

**Figure 4 | Comparison of theory and experimentally observed displacement fields.** The first row shows the theoretically computed Chladni figure[29], the second row shows the absolute displacement $|\mathbf{u}_n|$ and the third row shows the residual $\xi_n$. Scale bar (top left), 1 cm.

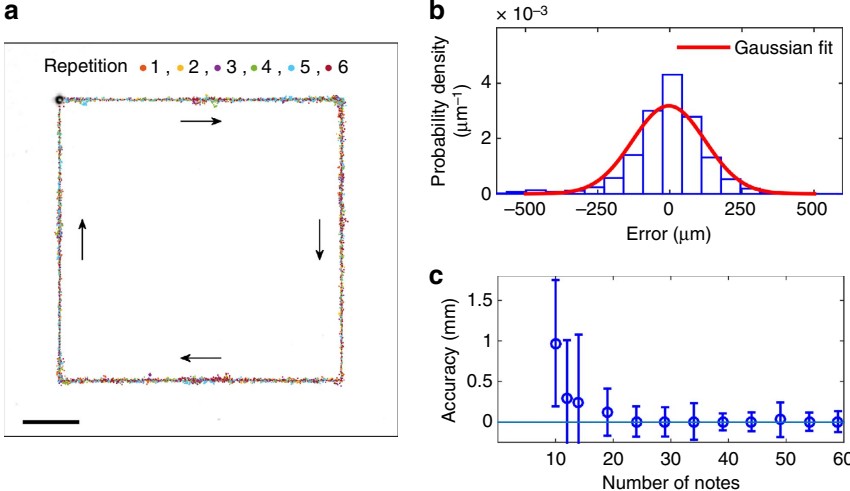

**Figure 5 | Manipulation accuracy and precision.** (**a**) Six repetitions of line tracking following a $25 \times 25$ mm box; scale bar, 5 mm; (**b**) An example error histogram against the reference line. (**c**) Line tracking accuracy versus the number of notes. The error bars represent s.d.

Peaks in the inverse of the nominal amplitude (Fig. 3b) are expected to correspond to the resonant peaks of the plate.

We use two alternative methods to find the resonances of the plate: acoustic characterization and theoretical computation[29]. Figure 3b compares the resonant peaks obtained using all three methods. The resonant peaks in inverse nominal amplitude and acoustic characterization agree well with each other, and the theory agrees with the two experimental methods with a similar accuracy as has been achieved before[29]. This confirms that there is a relationship between the resonances of the plate and the observed movement of the objects on the plate, which provides a quick way to estimate the nominal amplitude for each note from acoustic characterization or theory. On the other hand, it also shows that objects can be moved when the plate is not in resonance, if the plate can be actuated with sufficient power.

**Properties of the displacement field models**. The experimentally obtained displacement field can be related to theoretically predicted Chladni figures, with significant difference. Supplementary Fig. 1 shows that some of the vector fields, for example, $F^\#_6$, $A_6-F^\#_7$, $D_8-F_8$ and $G_8$, are clearly asymmetric, so

they cannot be constructed directly from symmetric theoretical models. The asymmetricity is attributed to the misalignment of the piezo actuator to the centre of the plate[30], where even very small misalignment can cause significant asymmetricity. On the other hand, the displacement field is not very sensitive to the mass on the plate where we noticed no obvious difference in the patterns for salt and solder balls.

The goodness of fit of our displacement field model can be assessed by comparing the magnitude of $\mathbf{u}_n$ to the magnitude of $\mathbf{e}$. To do this comparison, we fit a residual field function $\xi_n^2$ on the $(\mathbf{p}, |\mathbf{e}|^2)$ data for each note $n$ using similar LOESS-regression as before (see section Displacement fields in Methods). Figure 4 and Supplementary Fig. 1 show that the displacement $|\mathbf{u}_n|$ is typically small both at a node and an antinode, but $\xi_n$ is small at a node and large at an antinode. Supplementary Fig. 1 also shows that displacement $|\mathbf{u}_n|$ is mostly dominant compared with $\xi_n$ for notes between $F_6$ (1,397 Hz) and $E_9$ (10,548 Hz); however, this depends on the particular position on the plate.

**Manipulation precision**. We quantified the achievable control accuracy of the system by performing line tracking experiments

and measuring the tracking error from the reference line (Fig. 5a). The error histogram with all 59 notes in Fig. 5b shows close to a normal distribution. The measured data show a mean error of 4 μm and a s.d. of 125 μm. The mean error is below the resolution of our top view camera. The s.d. is equivalent to about 20% of the size of the 600 μm solder ball used in the test. Each pixel in our top view camera corresponds to ∼60 μm in the object space, which can already account for a significant part of the tracking error. We also quantified the smallest inducible particle motion using a side view high-speed camera, by progressively reducing signal duration. Supplementary Movie 9 shows an experiment where a single $F_8$ note (5,588 Hz) with a duration of 2.5 ms was played. The detected motion is one pixel in the side view, equivalent to 4 μm in object space. Therefore, the accuracy of our system could be further improved by increasing the top-view resolution from the current 60 μm and reducing the duration of the notes.

**Musical notes and manipulation.** In a centrally actuated Chladni plate, the nodal lines are points where the amplitude of a time-harmonic vibrating plate reaches its local minimum, occurring at all frequencies rather than at specific frequencies[29]. Therefore, the exact frequencies chosen for the manipulation are not particularly important, but rather the selected frequencies should span over a wide enough frequency range to produce a variety of Chladni figures on the plate. In our manipulation experiments, we used frequencies in western musical scale, ranging from $C_6$ (1.047 kHz) up to $A^{\#}_{10}$ (29.83 kHz), covering approximately the first seven theoretical resonant frequencies of the plate (Fig. 3b)[29]. The advantage of western musical scale is that each frequency has a convenient short-hand name; otherwise the frequencies are arbitrary and correspond to neither eigenfrequencies nor resonant frequencies of the plate.

The number of notes used in manipulation affects manipulation accuracy, number of time steps and if a trajectory can be followed. We performed manipulation experiments by using progressively fewer notes (Fig. 5c and Supplementary Fig. 4). For single-particle line tracking, 10 notes was the minimum number required to follow the square trajectory, and the tracking accuracy improves with more notes (Fig. 5c and Supplementary Fig. 4). The three particle manipulation (Fig. 1c) failed to reach the target in a reasonable time with 18 notes and succeeded with 34 notes, whereas the six particle manipulation (Fig. 2a) failed with 34 notes and succeeded with 59 notes. The sufficient number of notes depends on the complexity of the manipulation tasks, with more complex manipulation tasks requiring more notes. A theoretical lower bound can be derived from the theory of positive linear dependence[28]. The minimum number of basis vectors needed to positively span a space is $d + 1$, where $d$ is the dimension of the space. For our algorithm, $d = 2M$, where $M$ is the number of particles, because we control the two-dimensional displacement of each particle. Therefore, a necessary condition for the number of notes is $N \geq 2M + 1$, which is considerably smaller than the practically observed limit. We attribute this to the similarity between the vector fields of different notes.

**Avoiding proximity of objects during manipulation.** When two objects come too close to each other, their movement becomes coupled, and the controller may have difficulties in separating them. We have measured the minimum distance that the controller can separate two adjacent objects (Supplementary Fig. 5). The controller was 100% successful at separating two objects when the initial centre-to-centre distance was 2.5 mm, but the success rate dropped to 30% when the two objects started at a distance of 0.8 mm. The critical distance for the object motions to become coupled is related to the wavelength of the vibration. For

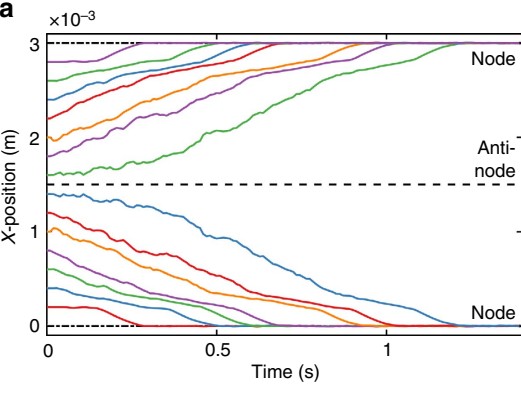

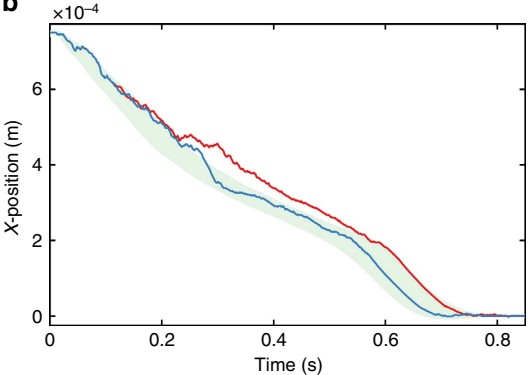

**Figure 6 | Results of object motion simulation.** (**a**) With varying initial position, the object is transported towards the nearest nodes. (**b**) Comparison of two simulation results with initial position shifted by 1 nm (solid red and blue lines). In 300 ms, the difference has grown to 100 μm, yet the positions meet again at the node. The shaded green area is the s.d.'s around the mean trajectory (not shown) of 100 simulations, when initial position was varied within a range of ±10 μm.

the highest manipulation notes $A^{\#}_{10}$ (29.83 kHz), the wavelength is 16 mm; for a one-dimensional wave, this means that the node–antinode distance is 4 mm, which corresponds well to the observed drop in the success rate. During the manipulation experiments, we avoid the coupling of motion by attempting to keep objects from becoming closer than 5 mm to each other, which is two times the distance at which the separation success rate started to drop. Below the threshold distance, the distance penalty is added to the cost function of the controller, which tries to move the objects away from each other (see sections Control algorithm and Manipulation experiments in Methods).

**Transport mechanism and simulations.** To understand the underlying transport mechanism, we observed the particle movement from side with a high-speed video camera (Supplementary Movie 10). The recording shows that the object spends much of its time in free flight between collisions with the plate. The collisions have a rocking pattern, where a collision on the left edge of the object is usually followed by a collision on the right edge (left and right as seen in the video). However, the rocking motion is not purely periodic: occasionally, collision on the left edge is followed by another collision on the same edge, and the durations between the collisions vary, without any definite pattern.

The object transport mechanism can be understood in terms of rigid body dynamics with impulse collisions[31]. We have developed a computer simulation model that is based on two-dimensional rigid body dynamics on a one-dimensional standing wave

(see Methods for details of the model). A simulated experiment is shown in Supplementary Movie 11. The object is transported towards the node and the quasi-periodic rocking motion is qualitatively similar to the experiment in Supplementary Movie 10. As expected, objects in general move towards the nearest node, finally settling at the node (Fig. 6a).

To check the sensitivity of the transport mechanism to initial conditions, we performed two simulation runs with only 1 nm difference in their initial position (red and blue curve in Fig. 6b). The system behaviour is chaotic: in 300 ms, a 1 nm change in initial position can lead up to 100 μm difference in the trajectories. The chaos is hardly surprising, because even simple models of a ball bouncing on a vibrating surface are known to be chaotic[32]. The chaotic nature of the object motion explains the difficulties in predicting the object motion accurately: small measurement errors in the initial position can lead to larger errors in the future. Nevertheless, the object motion in general is towards the nodes, that is, the nodes are attractors of this chaotic system[33].

To study the chaotic behaviour, we performed 100 simulation runs and varied the initial position within ± 10 μm, which is a small position measurement error in the scale of our system (50 × 50 mm plate), and computed the mean trajectory with s.d.'s (shaded green area in Fig. 6b). The results show that initially, the prediction error grows, yet as the objects come closer to the node, it starts to decrease again. Further simulation studies on the effects of various process parameters—including friction, restitution, frequency and dimensions—are given in Supplementary Note 1.

## Discussion

In this study, we show that the motion of multiple objects can be controlled simultaneously and independently using a Chladni plate with only a single actuator. Beyond the current apparatus, we anticipate our modelling and control method to enable more dexterous and parallel acoustic manipulation, which has numerous applications such as bottom-up cell culturing[1], lab-on-chips and microfluidics[3-6], cell and particle sorting[7,8], and patterning and characterization of bio-, micro- and nanomaterials[1,34]. We note that the underlying physical conversion mechanism from acoustic vibration into object motion is slightly different in different acoustic manipulation systems. Some use longitudinal vibrations of the media (sound)[35], while we use transverse vibrations of the plate. It is also well known that small particles (for example, flour) can move towards antinodes on a Chladni plate instead of nodes[36], an effect that can also be achieved by submerging Chladni membrane into water[37]. However, as long as repeatable motions are achieved and can be statistically modelled, our control method is agnostic to the physical origins of the model and thus can be applied. Furthermore, trapping objects at energetically stable points is a very general principle and widely applied to manipulation of objects in optical[38], electrostatic[39], magnetic[40] and acoustic fields[1-9], and Faraday waves[41]. Our modelling and control approach may have significant implications to all those methods: if we can apply fast enough feedback control compared with the motion of the objects, we should obtain a significantly greater number of degrees of freedom with the same actuation system, in both two- and three-dimensional applications, such as fluidic or air levitated systems.

## Methods

**Apparatus.** To make the plate, an unpolished silicon wafer (thickness 525 μm) was diced into a 50 × 50 mm square by using a mechanical wafer-dicing saw. A piezoelectric actuator (Piezomechanik/PSt 150/2 × 3/5, ~33 nm V$^{-1}$ displacement unloaded, −30 to 150 V) was mounted to the backside of the plate using cyanoacrylate adhesive. The piezoelectric actuator was mounted as centrally as possible by using jigs and visual guides. The plate is imaged by a camera (ImperX/IGV-B1621C-KC000 with Infinity/InfiniMite Alpha lens). Unpolished

plates were preferred because they did not produce any sharp reflections, making imaging more suitable for machine vision. Minor camera misalignment was corrected by digital roto-translation and cropping of the image; same image processing was applied consistently throughout the paper. Side illumination was provided by a ring of light-emitting diodes, mounted horizontally around and slightly above the plate. The side illumination provided good contrast, objects appearing bright on a dark background; however, throughout the paper, the colours in the micrographs were digitally inverted (dark on bright background) to improve readability in print. The camera was connected to a standard personal computer with a digital-to-analogue converter (National Instruments/USB-6363) for signal output. The output signal was amplified 20 × (Piezosystems/EPA-104-230) before sending it to the piezoelectric actuator. The voltages quoted in this paper are the voltages used to drive the actuator, after amplification, and amplitudes refer to the semi-amplitude (half peak-to-peak) voltage of a sinusoidal driving signal. The bias voltage of the driving signal was 60 V. Photograph of the experimental set-up is shown in Supplementary Fig. 6.

**Signal shape.** The notes played during manipulation were sinusoids with a triangular envelope, that is, $U = U_{peak}[1 - 2|(t - L/2)/L|] \sin(2\pi f t)$, $0 \leq t \leq L$, where $U$ is the voltage sent to the actuator, $U_{peak}$ is the amplitude of the signal, $t$ is time and $L$ is the duration of the signal and $f$ is the frequency of the played note. We used $L = 0.5$ s unless otherwise stated.

**Sample preparation and cleaning.** Before the experiments, the plate was cleaned using fibreless wipes wetted with acetone. By carefully dragging a wetted wipe over the plate at a constant speed, stain marks could be avoided. In the modelling and several manipulation experiments, the particles used were solder balls (Martin Smt/VD90.5106, Sb96.5Ag3Cu0.5, Ø 600 μm). To avoid excessive rolling of the balls on the plate, the sphericity of the solder balls was reduced by compressing them in a press. Scanning electron microscopy micrographs of the manipulated solder balls are shown in Supplementary Fig. 7; note the flats created by pressing. Before manipulation, the solder balls (Figs 1b,c, 2a and 3a) and the steel washer (outer diameter 7 mm, inner diameter 3 mm, Fig. 1d) were cleaned using acetone. The plant seeds of mustard and chia (sizes around 2 mm, Fig. 2b) were wiped with lens-cleaning tissues to remove dust. The candy ball (diameter around 1.6 mm, also known as nonpareil, Fig. 2b) was directly removed from original package. The radio frequency identification chips (730 × 730 × 70 μm) were directly removed from the dicing tapes (Supplementary Fig. 3). The SMT resistors (type 0805: 2.0 × 1.3 mm, Fig. 2b; type 0603: 1.5 × 0.8 mm) were directly removed from sealed packages. Water droplets (8 μl) were dispensed onto type 0805 SMT resistors using a manual liquid dispenser (Fig. 2c).

**Chladni figures.** To produce Chladni figures (Fig. 1b and Supplementary Fig. 8) on the plate, ordinary table salt was sprinkled on the plate and a continuous sinusoidal signal at a fixed frequency was played. The amplitude was increased until noticeable movement could be observed in the time frame of seconds and the experiment was continued until no more movement could be observed.

**Nominal amplitude.** The objective of the experiment was to find an amplitude for each note, which achieves, on average, the nominal displacement $s_{nom} = 175$ μm. Initially, solder balls (>100) were evenly distributed on the plate. During the experiment, 59 notes from the chromatic scale from $C_6$ (1.047 kHz) up to $A^{\#}_{10}$ (29.83 kHz) were played in a random order, every note 30 times (total of 59 × 30 = 1,770 experiments). Initially, the amplitude $U_{peak}$ for each note $n$ was set to 1.8 V. After at least every 50 notes, the particles were redistributed over the whole plate; playing the notes in random order ensured that the particles did not appreciably cluster during 50 time steps. The motion before and after playing a note was quantified by using machine vision; $s_{75\%} = 75\%$ quantile of $|\Delta\mathbf{p}|$ was used as a characteristic statistic for the motion of the particles. Before playing a note, we fitted a zero-intercept median regression model to the ($U_{peak}$, $s_{75\%}$) data for that note; we then used that model to choose the next $U_{peak}$ for that note such that the predicted displacement would be $s_{nom}$. Extrapolation was limited so that the newly chosen $U_{peak}$ would be at most 1.5 × the previous $U_{peak}$ that had $s_{75\%} < s_{nom}$ for that note. After ~20 adjustments (Supplementary Fig. 9), the amplitude $U_{peak}$ had converged, and the amplitude in the end of the experiment was taken as the nominal amplitude for each note (Fig. 3b). For five low-frequency notes between $E_6$ (1,319 Hz) and $G^{\#}_6$ (1,661 Hz), the amplitude was saturated at 90 V, which was the maximum driving voltage of the actuator, that is, even at maximum output, $s_{75\%}$ did not reach $s_{nom}$.

**Acoustic characterization.** The plate was excited with a logarithmic chirp signal (200 Hz–50 kHz, amplitude 20 V, duration 41 s), while recording the resulting sound with a pressure field microphone (Brüel & Kjær/4192, nominal frequency range 3.15–20 kHz, with 2669-C preamplifier and 2690-A signal conditioner, gain set to 10 mV Pa$^{-1}$). The microphone was placed at the centre of the plate, 10 mm above the plate surface. The measured signal was band-pass filtered with a Butterworth filter (pass band 0.6–40 kHz), squared and filtered using a Gaussian filter ($\sigma = 5$ ms) to estimate instantaneous output power, which is plotted as a function

of instantaneous frequency in Fig. 3b. Owing to practical limitations of the measurement set-up, the chirp experiment was conducted in 5 s segments and the whole spectrum was stitched from 10 measurements; each segment overlapping 1 s with the previous segment to avoid stitching artefacts. Note that the highest frequencies of the chirp signal are slightly above the nominal frequency range of the microphone; we did not observe appreciable loss of signal even at the higher frequencies. For converting pressures into decibels, the reference sound pressure was taken to be $20\,\mu$Pa.

**Resonance modelling.** For the theoretical estimation of the resonant frequencies and the resulting Chladni figures of the plate, we followed the approach of Tuan et al.[29], which also states that the resonance frequencies are different from eigenfrequencies in a centrally actuated plate. The method is based on solving the two-dimensional inhomogeneous Helmholtz equation. Briefly, the frequencies of resonant modes can be found using

$$f = CK^2 \qquad (2)$$

where $f$ is the frequency, $C$ is a constant dependent on the plate dimensions and material parameters, and $K$ is the wave number ($K = 2\pi/\lambda$, where $\lambda$ is the wavelength). $C$ can theoretically be estimated as

$$C_{\text{theor}} = \frac{1}{2\pi}\sqrt{\frac{Eh^2}{12\rho(1-v^2)}} \qquad (3)$$

where $E$ is the Young's modulus, $h$ is the thickness of the plate, $\rho$ is the density and $v$ is the Poisson ratio. Using nominal values[42] $E = 130$ GPa, $h = 525\,\mu$m, $\rho = 2{,}330$ kg m$^{-3}$ and $v = 0.28$, we found $C_{\text{theor}} \approx 0.188$ m$^2$ s$^{-1}$ for our plate. For a given plate shape, the resonant wave numbers $K$ can be found in the local maxima of the entropy function (Fig. 3b)[29]. To compute the entropy function, one needs the damping coefficient $\gamma$; however, the wave numbers of the resonant peaks do not depend heavily on the value of $\gamma$. We used $\gamma = 0.02w^{-1}$, where $w$ is the side length of the plate ($w = 50$ mm for our plate); this value has been shown to give valid results for a wide range of plates[29]. Experimentally, $C_{\text{exp}}$ was found by using the resonant wave numbers $K = 4.24, 5.14, 5.82, 6.16, 6.78$ and $7.62\pi w^{-1}$ and finding the corresponding frequency peaks in the acoustic power spectrum, based on $C_{\text{theor}}$ (Supplementary Fig. 10): $f = 11.9, 19.6, 23.9, 27.1, 34.5$ and $43.5$ kHz. Reason for only choosing wave numbers $K > 4\pi w^{-1}$ is that the method has only been validated in the high frequency limit ($K \gg \gamma$). We found $C_{\text{exp}}$ by fitting zero-intercept line to ($f^{1/2}$, $K$) data; this resulted in $C_{\text{exp}} \approx 0.184$ m$^2$ s$^{-1}$, which is close to the nominal value. Comparison of the theoretical and experimental dispersion curve is shown in Supplementary Fig. 11. Furthermore, we visually confirmed that the theoretical Chladni figures corresponded to the observed Chladni figures for each frequency (Supplementary Fig. 8). $C_{\text{exp}}$ was used in Fig. 3b to relate the wave numbers to frequency for the entropy function.

**Displacement fields.** Solder balls ($>98$, average 131) were evenly distributed over the whole plate and $N = 59$ notes from the chromatic scale from $C_6$ (1.047 kHz) up to $A^{\#}_{10}$ (29.83 kHz), were played in a random order, each note repeated 50 times. After at least every 50 notes, the particles were redistributed over the whole plate. The position $\mathbf{p} \in \mathbf{R}^2$ of each particle and displacement $\Delta\mathbf{p} \in \mathbf{R}^2$ after the note has been played were measured using machine vision. This resulted in a data set ($\mathbf{p}, \Delta\mathbf{p}$) for each note (number of data points $>6{,}323$, average 6,574). Data points with $|\Delta\mathbf{p}| > 3.5$ mm were discarded as erroneous matches by the machine vision. To reduce the computational load, the data set was compressed by binning. For each note, the plate was divided into a $30 \times 30$ grid and all data points within one grid cell were averaged. For each note $n \in 1..N$, we estimated $\mathbf{u}_n$ in equation (1) by performing robust LOESS-regression[27] on the binned values (span 7.5 mm). The name LOESS is derived from 'locally weighted scatterplot smoothing'. The main idea of LOESS is to use the values of nearby data points within the span to estimate the function value at any new point. For a given point $\mathbf{x}$, LOESS starts by fitting a second degree polynomial using weighted linear least-squares regression, based on data points within the span. The weights $W_i$ are given by

$$W_i = \left(1 - \left|\frac{\mathbf{x} - \mathbf{p}_i}{d(\mathbf{x})}\right|^3\right)^3 \qquad (4)$$

where $\mathbf{p}_i$ are the nearby data points of $\mathbf{x}$ within the span and $d(\mathbf{x})$ is the distance from $\mathbf{x}$ to the most distant data point within the span. The initial model is then made robust by discarding outlier data points, by assigning $W_i = 0$ for the outliers and updating weights for the remaining data points. In practice

$$W_i = \begin{cases} \left[1 - \left(\frac{r_i}{6\text{MAD}}\right)^2\right]^2, & |r_i| < 6\text{MAD} \\ 0, & |r_i| \geq 6\text{MAD} \end{cases} \qquad (5)$$

where $r_i$ is the residual of the $i$th data point produced by the regression smoothing procedure, and MAD is the median absolute deviation of the residuals. The above procedure was repeated five times, and the resulting final model was used to predict the value at the new point. We used Matlab implementation of LOESS.

Supplementary Fig. 1 shows all the resulting displacement fields. After fitting, we computed $|\mathbf{e}|^2 = |\Delta\mathbf{p} - \mathbf{u}_n(\mathbf{p})|^2$ for each data point and used similar binning and

fitting procedure on the ($\mathbf{p}, |\mathbf{e}|^2$) data to find the residual field $\xi_n^2: \mathbf{R}^2 \to \mathbf{R}$ for each note. To speed up computations during closed-loop control, the functions $\mathbf{u}_n$ and $\xi_n^2$ were stored in a $21 \times 21$ lookup table, which was linearly interpolated.

**Control algorithm.** The input to the control algorithm is an absolutely continuous cost function $J: \mathbf{P}^M \to \mathbf{R}$, where $\mathbf{P} = \mathbf{R}^2$ is the lateral position of an object on the plate and $M$ is the number of objects. The cost function should attain its minimum when the objects are at their desired positions, that is, it embodies the manipulation task. We use variants of root mean squared distances to the current target of each object, combined with additional penalty terms such as a penalty for two objects being too close to each other. At each time step $k$, the desired direction of movement $\mathbf{g}_k$ is computed as $\mathbf{g}_k = -\nabla J(x_{1,k}, y_{1,k}, ..., x_{m,k}, y_{m,k})$, where $x_{m,k}$ and $y_{m,k}$ are the coordinates of object $m \in 1..M$ at time $k$. $\mathbf{g}_k$ is a $2M \times 1$ vector. We then solve a linear programming problem:

$$\min_{\mathbf{w}_k}(\beta\mathbf{\Sigma}_k + (\beta-1)\mathbf{1})^T\mathbf{w}_k \text{ s.t. } \mathbf{\Psi}_k\mathbf{w}_k = \mathbf{g}_k,\ w_{n,k} \geq 0 \forall n \qquad (6)$$

where

$\beta \in [0, 1]$ is a scalar factor for tuning the controller,

$\mathbf{1}$ is a $N \times 1$ vector of ones,

$\mathbf{w}_k$ is a $N \times 1$ vector of the note weights with the elements $w_{n,k}$,

$\mathbf{\Sigma}_k = [\Sigma_{1,k}\ \ \Sigma_{2,k}\ \ \cdots\ \ \Sigma_{N,k}]^T$ is a $N \times 1$ vector with expected error for all notes at time $k$,

$\Sigma_{n,k} = \sum_{m=1}^{M}\xi_n^2(\mathbf{p}_{m,k})$ is a scalar value representing the total expected error for note $n$ at time $k$,

$\mathbf{p}_{m,k} = [x_{m,k}\ \ y_{m,k}]^T$ is the position of object $m$ at time $k$,

$\mathbf{\Psi}_k = [\psi_{1,k}\ \ \psi_{2,k}\ \ \cdots\ \ \psi_{N,k}]$ is a $2M \times N$ matrix with the modelled displacement fields for all the notes, sampled at the object positions, at time $k$,

$\psi_{n,k} = [\mathbf{u}_n^T(\mathbf{p}_{1,k})\ \ \mathbf{u}_n^T(\mathbf{p}_{2,k})\ \ \cdots\ \ \mathbf{u}_n^T(\mathbf{p}_{M,k})]^T$ is a $2M \times 1$ vector with the modelled displacement fields for note $n$ at time $k$, sampled at the object positions.

Small $\beta$ achieves faster, but stochastic motion; larger $\beta$ achieves slower, but less stochastic motion; we used $\beta = 0.2$. The condition $\mathbf{\Psi}_k\mathbf{w}_k = \mathbf{g}_k$ guarantees that the vector fields, when combined with the weights $\mathbf{w}_k$, achieve net movement in the desired direction of each object. The condition $w_{n,k} \geq 0$ guarantees that we never try to move opposite to the vector fields (away from the nodal line).

We then use $\mathbf{w}_k$ and stored previous weights to choose which note to play next. This is done using the following equations

$$b_{n,k} = (1 - a_{n,k-1})w_{n,k}^{-1}$$
$$n_k^* = \arg\min_n b_{n,k}$$
$$a_{n,k} = \begin{cases} a_{n,k-1} + w_{n,k}\min_{n'}b_{n',k} & , \text{if } n \neq n_k^* \\ 0 & , \text{otherwise} \end{cases} \qquad (7)$$

where $b_{n,k}$ is the relative weight remaining at time $k$ until a note $n$ will be played, $a_{n,k}$ is the accumulated weight of note $n$ at time $k$ ($a_{n,0} = 0\ \forall\ n$), and $n_k^*$ is the note that will be played at time $k$. It can be checked (proof omitted) that for constant weights, this conversion creates a sequence of notes where the relative occurrence of each note in the sequence is proportional to its weight. Finally, after having played note $n_k^*$, we adaptively update our plate model (the lookup tables for functions $\mathbf{u}_n$ and $\xi_n^2$) with the newly arrived data point.

**Manipulation experiments.** The trajectory following experiments (Fig. 1c,d and Supplementary Fig. 2) used 34 notes of the $C$-major scale from $C_6$ (1.047 kHz) up to $A_{10}$ (28.16 kHz), and sorting and pattern formation experiments (Fig. 2) used 59 notes of chromatic scale from $C_6$ (1.047 kHz) up to $A^{\#}_{10}$ (29.83 kHz). The amplitudes used in the manipulation were $1.1 \times$, $1.6 \times$, $0.9 \times$, $1.1 \times$ and $1.4 \times$ the nominal amplitude in Figs 1c,d and 2a–c, respectively. The duration $L$ of notes used was 0.5 s, except Fig. 2c, where $L = 0.25$ s was applied. In Fig. 1c,d, the cost function was root mean square of the distances of each particle to its waypoint. A new waypoint along the path was chosen when the object was closer than a threshold distance to its waypoint. Similarly, in Fig. 2a, the cost function was the root mean squared distance to the group centre. In Fig. 2b, the cost function was root mean square of the distances of particles to their corresponding targets in the pattern. In Fig. 2c, the cost function was the distance between the two droplets. To avoid clustering (Figs 1c and 2a,b), a penalty term of the form $\propto (\delta_{\text{thr}}\delta^{-1} - 1)$ was added, where $\delta$ is the distance of two particles and $\delta_{\text{thr}} = 5$ mm is the threshold distance.

**High-speed recording of particle–plate interaction.** A high-speed video camera (Phantom Miro M310) with a macro lens (Canon MP-E 65 mm f/2.8 1-5X Macro Photo) was installed on a tripod at one side of the apparatus to observe the particle–plate interaction, with a resolution of $640 \times 480$ pixels and frame rate of 10,000 FPS. Particles were placed on the plate at location around the point (1/4, 1/4) of plate size from the top-left corner. Three notes $C^{\#}_6$ (1,109 Hz), $A_6$ (1,760 Hz) and $F_8$ (5,588 Hz) having displacement fields parallel to the image plane of the camera at the location were selected to drive the particle.

                    

**Resolution and accuracy characterization.** The resolution of particle movement was determined using the high-speed camera while driving the piezo actuator with signals of small durations. The note used in the test was $F_8$ (5,588 Hz). We started the signal duration from the nominal 500 ms and reduced it gradually until no movement was observed. The smallest detected movement was 1 pixel, corresponding to about 4 μm in the object space, detected using a duration of 2.5 ms (Supplementary Movie 9).

Single-particle line tracking was conducted to characterize the tracking accuracy. A square of $25 \times 25$ mm was used as the reference trajectory, separated to multiple waypoints with 250 μm spacing. The accuracy was detected using a machine vision algorithm with the video camera in the apparatus, where each pixel in the camera corresponds to a $60 \times 60$ μm square on the plate. The tracking tests were repeated six times using the full 59 notes, with notes duration of 100 ms. A 50 μm threshold in machine vision reading was used by the controller to determine if a waypoint is reached. The tracking trajectory data points are down sampled 10 times for improved visualization in Fig. 5a.

**Trajectory following ability and number of notes.** To test the influence of the number of notes on trajectory following ability, the number of notes was linearly reduced in even spacing to the point where the experiment failed to complete in a reasonable time. A similar waypoint configuration was used as in the accuracy characterization, where a threshold was used by the controller to determine if a waypoint is reached. The waypoint spacing was 250 μm, except when the number of notes is equal to or below 14, where 1 mm spacing was used. The initial threshold was 50 μm; at 34 notes, the threshold was relaxed to 250 μm; and at 14 notes, the threshold was relaxed to 2.5 mm.

**Minimum separation distance.** Particle separation tests were carried out by placing two 600 μm solder balls randomly on the plate with a specific initial centre-to-centre distance and driving them 10 mm apart using the controller. Four initial distances were used: 0.8, 1, 2 and 2.5 mm. For each initial distance, we placed the two particles at 10 randomly generated positions on the plate, then the controller tried to separate them by driving them to opposite directions.

**Simulation.** The simulations in Fig. 6 and Supplementary Fig. 12 are based on two-dimensional rigid body dynamics with impulse collisions[31]. The object is perfectly rigid and shaped like a cylinder (diameter $D$ and height $H$), with axis in the plane of the two-dimensional simulation model. Initially, the centre of mass is at $(x_0, H/2)$, resting flat against the surface. The flight between collisions is purely ballistic, with gravitational acceleration $G$, neglecting hydrodynamics and drag.

Collisions are modelled using equations (39)–(49) from Wang and Mason[31]. The restitution model is based on Poisson's hypothesis, with a restitution coefficient of $e$. Poisson's hypothesis was preferred over Newton's law, which is known to occasionally violate energy conservation[31]. Dynamic friction and static Coulomb friction coefficients are set to the same value $\mu$.

The plate is modelled as an idealized one-dimensional standing wave, that is, the shape of the plate is modelled as $z(x, t) = A \sin(2\pi f t) \sin(2\pi x/\lambda)$, where $A$ is the amplitude, $f$ is the frequency, $t$ is the time, $\lambda$ is the wavelength and $x$ is the lateral position along the wave.

Only the corners of the object are considered as potential collision points, which is a valid assumption when the size of the object is small compared with the wavelength, that is, the curvature of the surface is negligible. Collision is detected when a corner is below the surface. In the rare occasion that two corners are detected to collide simultaneously, middle point between the two points is taken as the collision point.

Collisions with the plate are modelled as rigid wall collisions. For larger objects, this may be a source of error. With heavy particles, the particle collisions should dampen the vibrations of the plate. The practical consequence of this assumption is that the simulator predicts unrealistically large movements for larger objects. With the rigid wall assumption, mass of the object $m_o$ can be eliminated from collision equations, because the collision impulse $I$ is proportional to $m_o$ and the change velocity and angular velocity are proportional to $m_o^{-1} I$. Thus, the density of the material should have no effect on the numerical solution; however, changing mass in the simulator might lead to minor numerical rounding errors, which can be amplified by the chaotic behaviour of the system.

The simulator takes fixed time steps of $\Delta t$. When a collision is detected, no attempt is made to backtrack to the instant when the collision actually occurred, that is, the object may have already slightly penetrated the surface. Instead, we chose $\Delta t = f^{-1}/64$ so that within every vibration cycle, we detect the collision time with reasonable accuracy. For the purposes of computing the collision impulse[31], the surface tangent is taken as $(1, z_x(x_c, t_c))$, where $z_x$ is the spatial derivative of $z$, $x_c$ is the $x$-coordinate of the collision point and $t_c$ is the time when the collision was detected. The collisions are solved in the moving reference frame of the surface, where the surface velocity is $z_t(x_c, t_c)$, $z_t$ is time-based derivative of $z$.

Nominal values used in the simulation unless otherwise noted: $D = 600$ μm; $H = 300$ μm; $\lambda = 6$ mm; $x_0 = 750$ μm; $f = 1,000$ Hz; $A = 2.5$ μm; $G = 9.81$ ms$^{-2}$; $\mu = 0.5$; and $e = 0.5$.

**Data availability.** The data and the code that support the findings of this study are available from the corresponding author on request.

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

## Acknowledgements

V.S. was supported by Jenny and Antti Wihuri Foundation, Walter Ahlström Foundation and Academy of Finland (grants 268685 and 292477). K.L. and V.L. were supported by Aalto doctoral school of Electrical Engineering and Academy of Finland (grant 295006). We thank Prof. Ville Pulkki for the microphone used in the acoustic characterization and his insightful comments on the draft of this manuscript.

## Author contributions

Q.Z., V.S. and K.L. contributed equally to this work; Q.Z. and V.S. designed the research; K.L., Q.Z. and V.S. performed the experiments; V.S., K.L. and Q.Z. analysed the data; V.L. built the experimental platform; Q.Z. supervised the research. All authors wrote the manuscript and agreed on its final contents.

## Additional information

**Competing financial interests:** The authors declare no competing financial interests.

