## [Peer Review File · Nature Communications]

NCOMMS-16-07995-T

Reviewers' comments:

Reviewer #1 (Remarks to the Author):

This paper describes a new twist on a very old result - the Chladni plate experiment, which is taught to undergraduate engineers and physicists around the world. The present day authors have noted that each resonant mode (a.k.a. Chladni figures) produces a distinct pattern of surface displacements, or more specifically particle displacements. In their experiment they cover a plate with small particles and record their motions at each of the resonant modes. This then gives a large set of known displacements, which they call a model. This model is then wrapped into a control algorithm with vision as the feedback. For example, if it is desired to move a particle on the plate (in a particular position) to the left, then the algorithm chooses a vibration mode that best achieves this. They have demonstrated this experimentally and shown that the approach can be applied to multiple particles (6 is the maximum that is shown). The level of control of a single sub-mm (i.e. 600micron lead spheres) particle is good, but appears to deteriorate with more particles. Excellent control is achieved over a 7mm washer.

The paper represents a very nice proof of principle of a new type of positioning device. One very attractive feature is that the control is achieved with a single actuator. This is a dramatic simplification over previous attempts to control micro-particles which typically use many actuators.

Range of objects manipulated (materials and size). The present paper shows lead spheres (with flat regions) and a large washer. Neither of these is particularly useful in its own right, and neither can be said to even represent anything useful (that I can think of). I see 2 options: 1) show that something such as a cell, or an object that readers could recognise as representing a cell could be moved; 2) show that the approach is generally applicable by demonstrating the manipulation of a wider range of objects. In both cases I think the focus should be on the smaller scale objects as there are many other ways of manipulating larger objects such as washers.

In relation to the larger objects the authors should discuss, for example, the extensive work on linear motors and actuators, an example being: K. Sakano et al. Advanced Robotics 24 (2010) 1407-1421 which can be found at <http://www.kurosawa.ip.titech.ac.jp/publications/papers/jrsj2010k s.pdf>

The manipulation accuracy should be quantified. For example, it is apparent from the videos that there is significant 'jitter', this needs to be quantified. A key question is "is it random or systematic error?" At present the results look very encouraging, but without proper quantitative characterisation it is hard to compare this approach with other techniques.

In the abstract the authors suggest that the applications of this approach are in cell culturing and lab-on-a-chip devices. In all cases, the cells are required to be kept in cell culture medium, which is basically water. Will the present technique work underwater?

How do the displacement patterns found experimentally (e.g. Fig 1b) compare to theoretical prediction of the plate displacement? An alternative to the time consuming experimental approach used by the authors would be to use a simple model to predict the displacements.

Particle proximity is briefly discussed. Were the numbers quoted obtained experimentally? The authors should be more precise and perform an experiment specifically designed to probe this parameter.

The displacements are mapped onto a 30x30 grid. Can the authors comment on what happens if this grid is altered (both finer and coarser).

Reviewer #2 (Remarks to the Author):

This is an interesting system identification and feedback control approach to utilizing Chladni patterns on a vibrating plate to manipulate several small objects simultaneously in a horizontal plane. This is in contrast to cited work which uses more actuators to accomplish similar purposes. (The literature review misses work by Reznik and Canny at UC Berkeley which uses four actuators and a feedback control method to control multiple sliding parts simultaneously.)

The control method minimizing uncertainty while achieving the desired particle motion appears to be novel and interesting. The experimental demonstration videos provide a convincing proof of concept

that one actuator can control multiple objects simultaneously, though the motions of the objects are quite slow.

Drawbacks of this paper are:

1) There is no analysis of the transport mechanism causing parts to move when they are away from nodal lines. The fitted vector fields at each frequency are purely empirical, based on vision data from 50 experiments at each of 59 different frequencies with ~ 100 particles spread over the plate. The empirical model represents stochasticity by assigning a total variance in the predicted motion as a function of the location of the particle on the plate and the frequency of vibration. While this purely empirical approach is mentioned as a feature (there is little dependence on the specifics of the dynamics, other than that there must be a trend in the data beyond simply noise, allowing the approach to potentially be applied to other types of systems), the paper's primary focus is (a) Chladni-based manipulation, not (b) the generality of the modeling and control method. If the focus were (b), one would expect to see the approach applied to different types of systems. Since the focus is (a), one would like to see some explanation of the transport mechanism. Transport is likely primarily caused by repeated impacts with the vibrating surface, with nonzero friction and restitution coefficients. What is the role of the particle size, mass, friction and restitution coefficients, aspect ratio, and vibration frequency in generating transport and contributing to stochasticity? These could be addressed by a simplified model and/or by simulations. There is a minimal attempt to address these issues empirically by manipulating two types of objects (solder balls and a washer). Feedback control is expected to compensate for differences in particle properties.

2) The paper specifies the frequencies of the driving waveforms in both Hz and musical notation (e.g., C₆), but in general musical notes do not exactly correspond to the eigenfrequencies for a given plate. The paper should be clear on this. Is the method using frequencies of musical notes, or frequencies where Chladni patterns are predicted on the plate? If the former: why? The paper refers to "nodal lines" at all notes, when in fact nodal lines only exactly occur at specific frequencies.

3) Transport by repeated impacts may change the shape of the objects being manipulated, or damage them. Some modeling of the transport mechanism (see item 1 above) would help to address situations in which the approach is viable.

4) This method of manipulation appears to be quite slow. The videos are all significantly sped up. This is due to the stochasticity of the fields and the relatively weak "signal" in the noise. Also, the speed of manipulation slows as the number of particles increases.

5) While the extended data include nice plots of the empirical vector fields, there are no corresponding plots of the variance (e.g., a colormap as a function of position, where the color corresponds to the scalar variance). If all (or representative) vector fields and variance plots were normalized to the same scales, they would clarify the relative speeds of motion at the different frequencies and the amount of uncertainty.

6) The methods could be more succinctly and carefully explained. For example,

- The "triangular envelope" applied to sinusoids could be expressed as an equation, or otherwise explained more carefully.

- The functional form of the LOESS fit should be given.

- "statistically close to a nominal value of 175 μm " should be made precise.

- The meaning of "failed with 18 notes and succeeded with 34 notes" should be made precise. Presumably the particles could be moved closer to their goal states even with the smaller number of notes, but the form of the controller (requiring exact fitting to g) prevents a solution.

- The number of data points for each note should be clarified (e.g., approx 130 particles x 50 repetitions = 6500).

- The last page introduces the previously unused notations \hat{s}_n and \hat{v}_n .

Other items:

- Figure 3A is not strong evidence of superlinearity of the displacement with respect to amplitude. The data may be better described as demonstrating a "deadzone" where little motion occurs.

- Some videos could be improved by showing goal states before the particle motions.

- There appears to be significant asymmetry in the velocity fields, relative to the nodal patterns. Why?

- Chladni figures are generally analyzed with the assumption of nothing on the plate. If the mass of the objects on the plate is significant relative to the plate mass, the patterns will be affected. Some comment on this is warranted.

Reviewer #3 (Remarks to the Author):

This manuscript presents an algorithm-based control method that constructs note sequences acting on a Chladni plate in order to control the positions of multiple objects using a single acoustic actuator. It claims that: 1) the method can manipulate multiple objects simultaneously and independently, and be able to achieve control on more degrees of freedom than the recent demonstration using 64 actuators; 2) the method will enable more dexterous and parallel acoustic manipulation in cell culturing, lab-on-chips and microfluidics, cell and particle sorting, and patterning and characterization of bio-, micro and nanomaterials; 3) the method has significant implications to manipulation technologies of optical-based, electrostatic-based, magnetic-based and acoustic-based.

To manipulate objects independently and simultaneously is a very challenge task in acoustic-based manipulation due to the difficulty in isolation of specific acoustic fields to control the positions of objects independently. Some work has been conducted to overcome the challenge. Peng Zhang and Xiang Zhang et al. (Generation of acoustic self-bending and bottle beams by phase engineering, *Nature communication*, 2014) used phased engineering method to generate acoustic self-bending and bottle beams which can trap a single object and move it in demand. Asier Marzo and Sriram Subramanian et al. (Holographic acoustic elements for manipulation of levitated objects, *Nature communication*, 2015) optimized the phase array of acoustic elements that can construct certain acoustic fields which can levitate, translate, rotate, and rapid trap object. The idea presented in this manuscript is different from the previous works. It applies statistic tool and control algorithm to manipulate objects in demand based on elemental Chladni figures.

Although the idea is interesting, the results and data shown are not sufficient, and cannot convince the claims. Therefore, it is recommended to reject the manuscript. Further resubmission can be considered if the authors can provide more evidence to support their claims. The detail comments are given as following.

1. One of the key points of this manuscript is the model that directs the setup to move objects to target positions. However, it is not clearly stated. The definition of u_n ("a model fitted for note n ") is quite vague. It is difficult for the readers to understand how the algorithm can predict the next movement of object based on the current positions. Also, it is unclear how the algorithm realize such movement. To make it clear, it is better to give an example, which demonstrates how the model and

algorithm work in details, like moving along a straight line in at least 3 steps. Further, the mechanism of moving objects in Chladni plate is not discussed. What is the size limit of objects that can be manipulated on Chladni plate? Why is the model still suitable for a large 7-mm-diameter washer? According to the context, much more notes are required in practice than the theoretical prediction (for three particles, 34 notes, rather than 18; for six particles, 59 notes, rather than 34). Why are more notes needed in practice? What is the minimum number of notes to achieve successful manipulation in practice? What is the relation between notes number and notes duration? How do these two parameters affect the manipulation?

2. Based on the manuscript, it seems like that the manipulation of object in Chladni plate is achieved via linearly combined sequence notes which can produce basic Chladni figures. It is reasonable to use this method to manipulate a single object. However, for multiple objects, it is questionable that the same sequence notes can be used to independently and simultaneously control the movements. It is still unclear about why the independent and simultaneous control of multiple objects can be realized in this method. The movement of multiple objects in the same acoustic field can be different because of the different initial locations. Even though the experimental results in Fig. 1c and Fig. 2 seem like that the objects move independently, it is quite possibly a coincidence that the moving trajectories of the three particles just matched with the simple letters "S C I". In order to eliminate the concern of coincidence, it is suggested to demonstrate moving multiple objects independently and simultaneously along much more complicated traces, like the one shown in Fig. 1D. Also, for the results show in Fig. 2B, the final positions of the six objects can be matched by a circle or a rectangular, rather than a triangle it stated in the manuscript. To avoid misunderstanding, it is suggested to use more objects in the experiment for indicating the nodal lines more clearly.

3. The resolution of position control is not clear. From the result shown in Fig. 1 and Fig. 2, it looks like the error is quite large for objects in size of 600 μm . In addition, as stated in the manuscript, the movement of objects can be coupled together when they become close to each other ($<5\text{ mm}$). Actually, comparing to the size of objects (600 μm), the distance threshold is quite large, such that the precision of the position control in this method is questionable. The resolution of position control in this method should be given and proved via experiments. Moreover, manipulation of cells in lab-on-chip system and microfluidics requires much higher resolutions (1~10 μm). To support the claim 2 that the method can be useful in micro-scale manipulation, like cell culturing, lab-on-chips and microfluidics, it needs to demonstrate the ability of this method experimentally, at least to show that the method is able to manipulate objects in cell size independently and precisely.

Response to Referees for NCOMMS-16-07995-T

We thank the reviewers for their helpful suggestions and comments. We have addressed all the concerns raised by the reviewers. As the result, the paper has been substantially revised, including additional data, clarification of issues, extended discussion, and 6 new figures, 7 new videos, two updated figures and three updated videos. The major changes are:

1. We have applied the method to a wide range of miniature objects, including plant seeds, electronic components, candy balls, metal parts, and water droplets loaded on solid carriers. The sizes range from submillimetre (RFID chips, solder balls) to about 2 millimetres (mustard seeds, chia seeds, surface mount resistors, and candy balls), as well as 8 μ l droplets (see the updated Fig. 2, Extended Data Fig. 12, Video S6, S7, S10).
2. The transport mechanism has been studied using a high-speed camera, including two new Videos S8 and S11, as well as related discussion in the main text and Methods. A simulation model of the transport mechanism has been added: results in the later part of the main text and the new Fig. 4; a new section Simulation in the Methods; and an Extended discussion at the end of the paper, together with new Extended Data Fig. 13.
3. Properties of the method have been characterized, including the tracking accuracy (Extended Data Fig. 9), relation between number of notes and tracking ability (Extended Data Fig. 10), and minimum separation distance between two particles (Extended Data Fig. 11).
4. Several other updates including the addition of amplitude and variance map in the Extended Data Fig. 1, and moving of the perspective part of the abstract to the discussion.

In the following, we address the reviewers' comments point by point, where our responses are in bold.

Reviewer #1:

This paper describes a new twist on a very old result - the Chaladni plate experiment, which is taught to undergraduate engineers and physicists around the world. The present day authors have noted that each resonant mode (a.k.a. Chaladni figures) produces a distinct pattern of surface displacements, or more specifically particle displacements. In their experiment they cover a plate with small particles and record their motions at each of the resonant modes. This then gives a large set of known displacements, which they call a model. This model is then wrapped into a control algorithm with vision as the feedback. For example, if it is desired to move a particle on the plate (in a particular position) to the left, then the algorithm chooses a vibration mode that best achieves this. They have demonstrated this experimentally and shown that the approach can be applied to multiple particles (6 is the maximum that is shown). The level of control of a single sub-mm (i.e. 600micron lead spheres) particle is good, but appears to deteriorate with more particles. Excellent control is achieved over a 7mm washer.

The paper represents a very nice proof of principle of a new type of positioning device. One very attractive feature is that the control is achieved with a single actuator. This is a dramatic simplification over previous attempts to control micro-particles which typically use many actuators.

We thank the reviewer for understanding the novelty and significance of the work. A minor clarification: the frequencies exciting the plate in this paper are not necessarily resonance modes, but frequencies between 1047 Hz – 29834 Hz spaced in chromatic scale according to music theory.

1. Range of objects manipulated (materials and size). The present paper shows lead spheres (with flat regions) and a large washer. Neither of these is particularly useful in its own right, and neither can be said to even represent anything useful (that I can think of). I see 2 options: 1) show that something such as a cell, or an object that readers could recognise as representing a cell could be moved; 2) show that the approach is generally applicable by demonstrating the manipulation of a wider range of objects. In both cases I think the focus should be on the smaller scale objects as there are many other ways of manipulating larger objects such as washers.

We have made additional experiments to show that the approach is generally applicable: including 2 types of plant seeds (mustard, chia), 2 types of electronic components (RFID chips, surface mount resistor), candy balls, and water droplets.

We have added:

- **A short paragraph in the Abstract (lines 24-26):**

“Additionally, our method has been applied to manipulate a wide range of miniature objects, including electronic components, water droplets loaded on solid carriers, plant seeds, candy balls, and metal parts”.

- **A paragraph in the main text (lines 88-94):**

“We have also demonstrated pattern transformation, where we transform a diamond shape, defined by the four corner points, into a square shape (Fig. 2B and Movie S6). Four different types of miniature objects are used: a mustard seed, a chia seed, a candy ball, and a surface mount resistor. Besides solid objects, we have also transported water droplets on solid carriers (Fig. 2C and Movie S7). Two carriers each loaded with an 8 μ l water droplet were transported over a distance and finally merged into a single droplet. Finally, we have demonstrated aligning six radio frequency identification (RFID) chips in a line from initially scattered formation (Extended Data Fig. 12).”

2. In relation to the larger objects the authors should discuss, for example, the extensive work on linear motors and actuators, an example being: K. Sakano et al. *Advanced Robotics* 24 (2010) 1407-1421 which can be found at <http://www.kurosawa.ip.titech.ac.jp/publications/papers/jrsj2010ks.pdf>

Thank you for the comment. We have added discussion and citations to linear motors and actuators in the introduction (lines 35-36).

3. The manipulation accuracy should be quantified. For example, it is apparent from the videos that there is significant 'jitter', this needs to be quantified. A key question is "is it random or systematic error?" At present the results look very encouraging, but without proper quantitative characterisation it is hard to compare this approach with other techniques.

We have carried out experiments to characterize the manipulation accuracy (Extended Data Fig. 9). The accuracy for line tracking of a square shape is 4 μ m and the standard deviation is 125 μ m, where the pixel resolution of the camera in the object space (60 μ m) has a major contribution to the error. The jitter mainly comes from the randomness of the process, which is captured by our statistical model. We also noticed that it is more difficult to control in certain locations than

other locations due to the lacking of optimal vector fields towards the target location. This is reflected in the error histogram of Extended Data Fig. 9, where the left side tail deviates from normal distribution. In summary, the error is largely random, with local anomalies. The motion accuracy is however also affected by the camera resolution. The smallest incremental motion detected by high speed camera and microscope is 1 pixel, corresponding to around 4 μm (Video S11).

- We have added a paragraph in the Main Text (lines 144-154):

“We quantified the achievable control accuracy of the system by performing line tracking experiments and measuring the tracking error from the reference line. The error histogram with all 59 notes in Extended Data Fig. 9 shows close to a normal distribution. The measured data show an accuracy of 4 μm and a standard deviation of 125 μm . The standard deviation is equivalent to about 20 % of the size of the 600 μm solder ball used in the test. Each pixel in our top view camera corresponds to approximately 60 μm in the object space, which can already account for a significant part of the tracking error. We also quantified the smallest inducible particle motion using a side view high-speed camera, by progressively reducing signal duration. Video S11 shows an experiment where a single F8 note (5588Hz) with a duration of 2.5 ms was played. The detected motion is one pixel, equivalent to 4 μm in object space. Therefore, the accuracy of our system could be further improved by increasing the top view resolution from the current 60 μm and reducing the duration of the notes.”

- We have also added a section “Resolution and accuracy characterization” in the Methods (lines 436-448).

4. In the abstract the authors suggest that the applications of this approach are in cell culturing and lab-on-a-chip devices. In all cases, the cells are required to be kept in cell culture medium, which is basically water. Will the present technique work underwater?

We apologize for the poor wording. The last section of the abstract was not the summary of our work, but the future perspective of the work according to the original submission format. To make things clear, we have moved the last section of the abstract to the end of the paper.

The present apparatus will not work underwater. However, the modelling and control methods should be applicable to other apparatus having nodal lines in liquid¹, which we also plan to do in the future. On the other hand, we have also demonstrated transportation of water droplets on solid carriers, which has potential applications in lab-on-chip applications.

- We have added more detailed discussion on potential applicability of our method in the main text (lines 197-210):

“Beyond the current apparatus, we anticipate our modelling and control method to enable more dexterous and parallel acoustic manipulation, which has numerous applications such as bottom-up cell culturing, lab-on-chips and microfluidics, cell and particle sorting, and patterning and characterization of bio-, micro- and nanomaterials. We note that the underlying physical conversion mechanism from acoustic vibration into object motion is slightly different in different acoustic manipulation systems. Some use longitudinal vibrations of the media (sound), while we use transverse vibrations of the plate. It is also well known that small particles (e.g. flour) can move towards antinodes on a Chladni plate instead of nodes, an effect that can also be achieved by submerging Chladni membrane into water. However, as long as repeatable motions are achieved and can be statistically modelled, our

control method is agnostic to the physical origins of the model and thus can be applied. Furthermore, trapping objects at energetically stable points is a very general principle and widely applied to manipulation of objects in optical, electrostatic, magnetic and acoustic fields and Faraday waves.”

5. How do the displacement patterns found experimentally (e.g. Fig 1b) compare to theoretical prediction of the plate displacement? An alternative to the time consuming experimental approach used by the authors would be to use a simple model to predict the displacements.

As shown in the Extended Data Fig. 1, some of the patterns are clearly asymmetric, so they cannot be constructed directly from the symmetric theoretical model. The asymmetry is attributed to the misalignment of the piezo actuator to the centre of the plate, where even very small misalignment can cause significant asymmetry². Additionally, for the largely symmetric patterns, the velocity profile diverts from the simple amplitude gradient of the theoretical model due to the chaotic motion of the particle on the plate (Fig. 4, Video S8, S9). Therefore, it is hard to derive a theoretical model to predict the particle displacement across the plate in an accuracy sufficient for manipulation.

- **We have added to the main text (lines 110-115):**

“The experimentally obtained displacement field can be related to theoretically predicted Chladni figures, with significant difference. Extended Data Figure 1 shows that some of the vector fields e.g. F#6, A6-F#7, D8-F8 and G8, are clearly asymmetric, so they cannot be constructed directly from symmetric theoretical models. The asymmetry is attributed to the misalignment of the piezo actuator to the centre of the plate, where even very small misalignment can cause significant asymmetry.”

6. Particle proximity is briefly discussed. Were the numbers quoted obtained experimentally? The authors should be more precise and perform an experiment specifically designed to probe this parameter.

We have conducted new experiments to check the influence of particle proximity at different initial centre-to-centre distances of 0.8 mm, 1.0 mm, 2.0 mm and 2.5 mm. For each initial distance, we placed two particles at 10 randomly generated positions on the plate, then the controller tries to separate them by increasing the distance to 10 mm. The results show that the separation is 100% successful with initial distance of 2.5 mm (Extended Data Fig. 11).

- **A paragraph has been added in main text (lines 156-159):**

“We have measured the minimum distance that the controller can separate two adjacent objects (Extended Data Fig. 11). The controller was 100% successful at separating two objects when the initial centre-to-centre distance was 2.5 mm, but the success rate dropped to 30% when the two objects started at a distance of 0.8 mm.”

7. The displacements are mapped onto a 30x30 grid. Can the authors comment on what happens if this grid is altered (both finer and coarser).

The number of cells in the grid should be sufficient to capture the variations in the vector fields. In the Extended Data Fig. 1, we can find the maximum number of peaks in one axis is 6 at the maximum frequency we used (29834 Hz), so 30x30 is a reasonable size. We have tested 800x800

grid and did not observe significant difference in the performance. According to Nyquist-Shannon sampling theorem, 12×12 is the minimum number of grids to capture the variations. In practice, more is normally recommended.

Reviewer #2:

This is an interesting system identification and feedback control approach to utilizing Chladni patterns on a vibrating plate to manipulate several small objects simultaneously in a horizontal plane. This is in contrast to cited work which uses more actuators to accomplish similar purposes. (The literature review misses work by Reznik and Canny at UC Berkeley which uses four actuators and a feedback control method to control multiple sliding parts simultaneously.)

The control method minimizing uncertainty while achieving the desired particle motion appears to be novel and interesting. The experimental demonstration videos provide a convincing proof of concept that one actuator can control multiple objects simultaneously, though the motions of the objects are quite slow.

We thank the reviewer for all the comments. We have added the citation to the work of Reznik and Canny. The current work is a proof of concept; we will continue work on the research to improve the speed in the future with better engineering and control system with higher sampling rate.

Drawbacks of this paper are:

1. There is no analysis of the transport mechanism causing parts to move when they are away from nodal lines. The fitted vector fields at each frequency are purely empirical, based on vision data from 50 experiments at each of 59 different frequencies with ~ 100 particles spread over the plate. The empirical model represents stochasticity by assigning a total variance in the predicted motion as a function of the location of the particle on the plate and the frequency of vibration. While this purely empirical approach is mentioned as a feature (there is little dependence on the specifics of the dynamics, other than that there must be a trend in the data beyond simply noise, allowing the approach to potentially be applied to other types of systems), the paper's primary focus is (a) Chladni-based manipulation, not (b) the generality of the modeling and control method. If the focus were (b), one would expect to see the approach applied to different types of systems. Since the focus is (a), one would like to see some explanation of the transport mechanism. Transport is likely primarily caused by repeated impacts with the vibrating surface, with nonzero friction and restitution coefficients. What is the role of the particle size, mass, friction and restitution coefficients, aspect ratio, and vibration frequency in generating transport and contributing to stochasticity? These could be addressed by a simplified model and/or by simulations. There is a minimal attempt to address these issues empirically by manipulating two types of objects (solder balls and a washer). Feedback control is expected to compensate for differences in particle properties.

We thank the reviewer for the detailed comments. The paper has been significantly revised to address this comment. We have recorded the actual motion of the objects on the vibrating plate (Video S8), and built a simulation model to simulate the motion of the particle during vibration. A discussion section on simulation has been added to the later part of the main text, a new section "Simulation" has been added to the Methods, as well as an Extended Discussion to explain the potential influence of vibration amplitude, particle size, mass, aspect ratio, initial position,

friction coefficient, restitution coefficient and plate vibration frequency. We have also added two new figures (Fig. 4 and Extended Data Fig. 13) and one new video (Video S9).

For the type of objects, we have made additional experiments to show that the approach is generally applicable: including 2 types of plant seeds (mustard, chia), 2 types of electronic components (RFID chips, surface mount resistor), candy balls, and water droplets.

- We have added in the main text (lines 167-195):

“To understand the underlying transport mechanism, we observed the particle movement from side with a high-speed video camera (Movie S8). The recording shows that the object spends much of its time in free flight between collisions with the plate. The collisions have a rocking pattern, where a collision on the front edge of the motion is usually followed by a collision on the back edge. However, the rocking motion is not purely periodic: occasionally collision on the front edge is followed by another collision on the front edge and the durations between the collisions vary, without any definite pattern.

The object transport mechanism can be understood in terms of rigid body dynamics with impulse collisions. We have developed a computer simulation model that is based on two-dimensional rigid body dynamics on a one-dimensional standing wave (see Methods for details of the model). A simulated experiment is shown in Movie S9. The object is transported towards the node and the quasi-periodic rocking motion is qualitatively similar to the experiment in Movie S8. As expected, objects in general move towards the nearest node, finally settling at the node (Fig. 4A).

To check the sensitivity of the transport mechanism to initial conditions, we performed two simulation runs with only 1 nm difference in their initial position (red and blue curve in Fig. 4B). The system behaviour is chaotic: in 200 ms, a 1 nm change in initial position can lead up to 100 μm difference in the trajectories. The chaos is hardly surprising, because even simple models of a ball bouncing on a vibrating surface are known to be chaotic. The chaotic nature of the object motion explains the difficulties in predicting the object motion accurately: small measurement errors in the initial position can lead to larger errors in the future. Nevertheless, the object motion in general is towards the nodes i.e. the nodes are attractors of this chaotic system.

To study the chaotic behaviour, we performed 100 simulation runs and varied the initial position within $\pm 10 \mu\text{m}$, which is a small position measurement error in the scale of our system (50 \times 50 mm plate), and computed the mean trajectory with standard deviations (shaded green area in Fig. 4B). The results show that initially, the prediction error grows, yet as the objects come closer to the node, it starts to decrease again. Thus, accurate position predictions of the objects are only feasible either with a small time horizon (as in our method), or when waiting long enough that the object is close to a node (as Chladni did). Further simulation studies on the effects of various process parameters - including friction, restitution, frequency and dimensions - are given in Extended Discussion 1.”

- We have added a Simulation section in Methods (lines 463-497) and an Extended discussion (lines 691-728).
2. The paper specifies the frequencies of the driving waveforms in both Hz and musical notation (e.g., C₆), but in general musical notes do not exactly correspond to the eigenfrequencies for a given plate. The paper should be clear on this. Is the method using frequencies of musical notes, or frequencies

where Chladni patterns are predicted on the plate? If the former: why? The paper refers to "nodal lines" at all notes, when in fact nodal lines only exactly occur at specific frequencies.

We would like to clarify that the nodes are understood as points where the amplitude of a time-harmonic vibrating plate reaches its local minimum. In a centrally actuated plate, they occur at all frequencies rather than at specific frequencies³ (see also Extended Data Fig. 1). Additionally, resonant frequency is different from eigenfrequency in a centrally actuated plate³. In our experiment, we chose our frequencies such that they are logarithmically spaced, and one such spacing is the western musical scale, where the main benefit is the shorthand name for each frequency.

- **We have clarified this in the main text (lines 125-129)**

“In our manipulation experiments, we used frequencies in western musical scale, ranging from 1.047 kHz (musical note C6) up to 29.83 kHz (A#10), covering approximately the first seven theoretical resonant frequencies of the plate (Fig. 3B). The main advantage of western musical scale is that each frequency has a convenient short-hand name; otherwise the frequencies are arbitrary and correspond to neither eigenfrequencies nor resonant frequencies of the plate.”

- **We also clarified the section Resonance modelling in Methods (lines 322-324).**

3. Transport by repeated impacts may change the shape of the objects being manipulated, or damage them. Some modeling of the transport mechanism (see item 1 above) would help to address situations in which the approach is viable.

We have recorded the transport mechanism using a high-speed camera (see Video S8). There is no observation of damage to the parts under manipulation, even for relatively soft materials such as solder balls.

We have also simulated the transport mechanism, see the added discussion in the main text (lines 167-195), a new Simulation section in Methods (lines 463-497) and an Extended discussion (lines 691-728).

4. This method of manipulation appears to be quite slow. The videos are all significantly sped up. This is due to the stochasticity of the fields and the relatively weak "signal" in the noise. Also, the speed of manipulation slows as the number of particles increases.

We agree that the manipulation speed of the current apparatus is not fast. The current work focuses on the working principle to build a proof-of-concept apparatus. We plan to continuously refine different engineering aspects in a new project which is just starting: to develop new apparatus with improved system construction, faster programmable signal generator, and better machine vision to improve the performance.

5. While the extended data include nice plots of the empirical vector fields, there are no corresponding plots of the variance (e.g., a colormap as a function of position, where the color corresponds to the scalar variance). If all (or representative) vector fields and variance plots were normalized to the same scales, they would clarify the relative speeds of motion at the different frequencies and the amount of uncertainty.

A colormap of the amplitude and variance of the vector field has been added to the Extended Data Fig. 1, with normalized scale.

6. The methods could be more succinctly and carefully explained. For example,
- The "triangular envelope" applied to sinusoids could be expressed as an equation, or otherwise explained more carefully.

An equation has been added with a new section "Signal shape" in Methods (lines 268-271).

- The functional form of the LOESS fit should be given.

The LOESS cannot be expressed in a single equation, but we have added explanation of the algorithm with the key equations in functional form to the section "Displacement fields" in Methods (lines 359-374).

- "statistically close to a nominal value of 175 μm " should be made precise.

We have clarified it in the main text (lines 102-104):

"For modelling of the displacement field, we chose to keep the duration of the notes constant (0.5 s), and algorithmically adjusted the amplitude of each note such that the typical displacement for all notes is close to 175 μm (see section Nominal amplitude in Methods)."

- The meaning of "failed with 18 notes and succeeded with 34 notes" should be made precise. Presumably the particles could be moved closer to their goal states even with the smaller number of notes, but the form of the controller (requiring exact fitting to g) prevents a solution.

We thank the reviewer for pointing this out. With lower amount of notes, the controller was able to move the particles but cannot reach the target in a reasonable time, because the particles dance in certain patterns or around certain location. "Failed" means this. The reviewer is also right that the current form of controller is relatively greedy. An advanced path planning algorithm may lead to better results. The current paper is a proof-of-concept, and we try to apply an easy-to-understand controller. We will develop advanced controller and path planning algorithm in the future.

For the current controller, we have also explained the working principle in the new Video S2.

- **We have clarified in the main text for the failure (lines 130-136):**

"The number of notes used in manipulation affects manipulation accuracy, number of iterations, and if a trajectory can be followed. We performed manipulation experiments by using progressively fewer notes (Extended Data Fig. 10). For single particle line tracking, 10 notes was the minimum number required to follow the square trajectory, and the tracking accuracy improves with more notes (Extended Data Fig. 10). The three particle manipulation (Fig. 1C) failed to reach the target in a reasonable time with 18 notes and succeeded with 34 notes, whereas the six particle manipulation (Fig. 2A) failed with 34 notes and succeeded with 59 notes."

- The number of data points for each note should be clarified (e.g., approx 130 particles x 50 repetitions = 6500).

We have clarified in the main text for the number of notes (lines 52-54):

"The number of particles on the plate was 131 on average, with each note being played 50 times, resulting in approximately 6600 data points per note and 390,000 data points in total."

- The last page introduces the previously unused notations \hat{s}_n and \hat{v}_n .

Sorry for the typo, we have corrected it.

7. Other items:

- Figure 3A is not strong evidence of superlinearity of the displacement with respect to amplitude. The data may be better described as demonstrating a "deadzone" where little motion occurs.

We agree with the reviewer's comment. We have removed superlinearity claims and added in the main text (lines 97-98):

"The observed relationship with duration is roughly linear, while the relationship with amplitude is close to linear but having a dead zone."

- Some videos could be improved by showing goal states before the particle motions.

We have added goal states to Videos S3, S4, and the new added Videos S2, S7 and S10.

- There appears to be significant asymmetry in the velocity fields, relative to the nodal patterns.

Why?

The asymmetry is attributed to the misalignment of the piezo actuator to the centre of the plate, where even very small misalignment can cause significant asymmetry².

- **We have added to the main text (lines 110-115):**

"The experimentally obtained displacement field can be related to theoretically predicted Chladni figures, with significant difference. Extended Data Figure 1 shows that some of the vector fields e.g. F#6, A6-F#7, D8-F8 and G8, are clearly asymmetric, so they cannot be constructed directly from symmetric theoretical models. The asymmetry is attributed to the misalignment of the piezo actuator to the centre of the plate, where even very small misalignment can cause significant asymmetry."

- Chladni figures are generally analyzed with the assumption of nothing on the plate. If the mass of the objects on the plate is significant relative to the plate mass, the patterns will be affected. Some comment on this is warranted.

We are using a centrally actuated Chladni plate, where the actuator continuously inputs energy to the system and the vibration is less sensitive to the mass on the plate, as reported by Tuan et. al³. We also noticed that there was no difference in the patterns for salt and solder balls.

- **We have added the discussion in the main text (lines 115-118):**

"On the other hand, the displacement field is not very sensitive to the mass on the plate where we noticed no obvious difference in the patterns for salt and solder balls. For a centrally actuated Chladni plate, the vibration is less sensitive to the mass on the plate because the actuator continuously inputs energy into the system."

Reviewer #3:

This manuscript presents an algorithm-based control method that constructs note sequences acting on a Chladni plate in order to control the positions of multiple objects using a single acoustic actuator. It claims that: 1) the method can manipulate multiple objects simultaneously and independently, and be able to achieve control on more degrees of freedom than the recent demonstration using 64 actuators; 2) the method will enable more dexterous and parallel acoustic manipulation in cell culturing, lab-on-chips and microfluidics, cell and particle sorting, and patterning and characterization of bio-, micro and nanomaterials; 3) the method has significant implications to manipulation technologies of optical-based, electrostatic-based, magnetic-based and acoustic-based.

We thank the reviewer for all the comments. We apologize for the poor wording in the abstract. The last section of the abstract was not the summary of our work, but the future perspective of the work according to the guideline of the original submission. To make things clear, we have moved the last section of the abstract (covering point 2 and 3 mentioned by the reviewer) to the discussion at the end of the main text of paper.

To manipulate objects independently and simultaneously is a very challenge task in acoustic-based manipulation due to the difficulty in isolation of specific acoustic fields to control the positions of objects independently. Some work has been conducted to overcome the challenge. Peng Zhang and Xiang Zhang et al. (Generation of acoustic self-bending and bottle beams by phase engineering, Nature communication, 2014) used phased engineering method to generate acoustic self-bending and bottle beams which can trap a single object and move it in demand. Asier Marzo and Sriram Subramanian et al. (Holographic acoustic elements for manipulation of levitated objects, Nature communication, 2015) optimized the phase array of acoustic elements that can construct certain acoustic fields which can levitate, translate, rotate, and rapid trap object. The idea presented in this manuscript is different from the previous works. It applies statistic tool and control algorithm to manipulate objects in demand based on elemental Chladni figures.

We thank the reviewer for noticing the challenge of the task. We have added the citation to Zhang 2014.

Although the idea is interesting, the results and data shown are not sufficient, and cannot convince the claims. Therefore, it is recommended to reject the manuscript. Further resubmission can be considered if the authors can provide more evidence to support their claims. The detail comments are given as following.

1. One of the key points of this manuscript is the model that directs the setup to move objects to target positions. However, it is not clearly stated. The definition of un ("a model fitted for note n") is quite vague.

We have clarified in the main text (lines 57-58): “ u_n is a two-dimensional displacement model”.

It is difficult for the readers to understand how the algorithm can predict the next movement of object based on the current positions. Also, it is unclear how the algorithm realize such movement. To make it clear, it is better to give an example, which demonstrates how the model and algorithm work in details, like moving along a straight line in at least 3 steps.

We have added a video illustration in the new Video S2 to explain the control algorithm. We also clarified the main text (lines 64-74):

“We use the model to control the motion of the objects on the plate by iteratively generating a note sequence that, on average, moves the objects towards their desired directions. In each iteration, the computer finds a positive combination of all the modelled displacement fields (Extended Data Fig. 1) such that the net motion for all the objects at their current position is guided towards the desired direction. The positive combination is interpreted as a weight for each note. The computer then converts these weights into a sequence of notes, where the relative rate of occurrence of a note in the sequence is proportional to the weight of that note. For example, if in the first iteration notes A, B and C have the weights 3/6, 2/6 and 1/6, respectively, then the implied note sequence to be played is ABAABCABAABC...

corresponding to the weights. In practice, the note sequence is updated in every iteration based on the object locations and desired directions. The details of the control method are explained thoroughly in Methods, as well as in Movie S2.”

Further, the mechanism of moving objects in Chladni plate is not discussed. What is the size limit of objects that can be manipulated on Chladni plate? Why is the model still suitable for a large 7-mm-diameter washer?

The paper has been significantly revised to address this comment. We have recorded the actual motion of the objects on the vibrating plate (Video S8), and built a simulation model to simulate the motion of the particle during vibration. A discussion section on simulation has been added to the later part of the main text, a new section “Simulation” has been added to the Methods, as well as an Extended Discussion to explain the potential influence of vibration amplitude, particle size, mass, aspect ratio, initial position, friction coefficient, restitution coefficient and plate vibration frequency. We have also added two new figures (Fig. 4 and Extended Data Fig. 13) and one new video (Video S9).

For larger objects, when the contact points between the object and the plate spans a distance larger than the distance between nodes and antinodes, the control becomes more complicated, which is a limitation. For smaller objects, the limits come from aerodynamics, where different system behavior occurs at e.g. particle size about $75 \mu\text{m}^4$. In such cases, the vector models will be rather different, even though there is good chance our control methods should still apply. In our experiments for large object such as the washer, the direction of the vector field remains the same, but it requires more power to move. We used a voltage 1.6 times the nominal amplitude as input (see section “Manipulation experiments” in Methods).

- We have added in the main text (lines 167-195):

“To understand the underlying transport mechanism, we observed the particle movement from side with a high-speed video camera (Movie S8). The recording shows that the object spends much of its time in free flight between collisions with the plate. The collisions have a rocking pattern, where a collision on the front edge of the motion is usually followed by a collision on the back edge. However, the rocking motion is not purely periodic: occasionally collision on the front edge is followed by another collision on the front edge and the durations between the collisions vary, without any definite pattern.

The object transport mechanism can be understood in terms of rigid body dynamics with impulse collisions. We have developed a computer simulation model that is based on two-dimensional rigid body dynamics on a one-dimensional standing wave (see Methods for details of the model). A simulated experiment is shown in Movie S9. The object is transported towards the node and the quasi-periodic rocking motion is qualitatively similar to the experiment in Movie S8. As expected, objects in general move towards the nearest node, finally settling at the node (Fig. 4A).

To check the sensitivity of the transport mechanism to initial conditions, we performed two simulation runs with only 1 nm difference in their initial position (red and blue curve in Fig. 4B). The system behaviour is chaotic: in 200 ms, a 1 nm change in initial position can lead up to 100 μm difference in the trajectories. The chaos is hardly surprising, because even simple models of a ball bouncing on a vibrating surface are known to be chaotic. The chaotic nature of the object motion explains the difficulties in predicting the object motion accurately: small measurement errors in the initial position can lead to larger errors in the future.

Nevertheless, the object motion in general is towards the nodes i.e. the nodes are attractors of this chaotic system.

To study the chaotic behaviour, we performed 100 simulation runs and varied the initial position within $\pm 10 \mu\text{m}$, which is a small position measurement error in the scale of our system ($50 \times 50 \text{ mm}$ plate), and computed the mean trajectory with standard deviations (shaded green area in Fig. 4B). The results show that initially, the prediction error grows, yet as the objects come closer to the node, it starts to decrease again. Thus, accurate position predictions of the objects are only feasible either with a small time horizon (as in our method), or when waiting long enough that the object is close to a node (as Chladni did). Further simulation studies on the effects of various process parameters - including friction, restitution, frequency and dimensions - are given in Extended Discussion 1.”

- We have added a Simulation section in Methods (lines 463-497) and an Extended discussion (lines 691-728).

According to the context, much more notes are required in practice than the theoretical prediction (for three particles, 34 notes, rather than 18; for six particles, 59 notes, rather than 34). Why are more notes needed in practice?

What is the minimum number of notes to achieve successful manipulation in practice?

Theoretical prediction uses the ideal minimum number of notes where each note can produce desired local field for each particle. In practice, there is similarity of the vector fields produced by each note, and many more notes are required to create the desired motion for each particle depending on the actual field of each note.

We have carried out square trajectory following experiments where notes are dropped until square following fails. The minimum number of notes depends on the tolerance used by the controller. The absolute minimum is 10 notes to follow a square using a single particle.

- We added this discussion in the main text (lines 130-143):
“The number of notes used in manipulation affects manipulation accuracy, number of iterations, and if a trajectory can be followed. We performed manipulation experiments by using progressively fewer notes (Extended Data Fig. 10). For single particle line tracking, 10 notes was the minimum number required to follow the square trajectory, and the tracking accuracy improves with more notes (Extended Data Fig. 10). The three particle manipulation (Fig. 1C) failed to reach the target in a reasonable time with 18 notes and succeeded with 34 notes, whereas the six particle manipulation (Fig. 2A) failed with 34 notes and succeeded with 59 notes. The sufficient number of notes depends on the complexity of the manipulation tasks, with more complex manipulation tasks requiring more notes. A theoretical lower bound can be derived from the theory of positive linear dependence. The minimum number of basis vectors needed to positively span a space is $d+1$, where d is the dimension of the space. For our algorithm, $d=2M$, where M is the number of particles, because we control the two-dimensional displacement of each particle. Therefore, a necessary condition for the number of notes is $N \geq 2M+1$, which is considerably smaller than the practically observed limit. We attribute this to the similarity between the vector fields of different notes.”
- A new section “Trajectory following ability vs. number of notes” has been added to the Methods (line 449-456), as well as the new Extended Data Fig. 10.

What is the relation between notes number and notes duration? How do these two parameters affect the manipulation?

If by “notes number” the reviewer means the name of the notes, which determines the frequency, then the frequency and the duration of a note are independent parameters. Both frequency and duration affect the movement. Each frequency is associated with a displacement field (Extended Data Fig. 1). The notes duration has a close to linear relation with the amplitude of the displacement field (see Fig. 3A).

If by “notes number” the reviewer means the number of notes, then increasing the number of notes generally leads to better performance in manipulation (Extended Data Fig. 10). On the other hand, reducing the duration of notes leads to smaller steps, which may result in better accuracy if the number of notes are kept the same. The side effect is a larger number of required steps to cover a distance.

We added this discussion to the main text (lines 130-143), see the response to the previous comment.

2. Based on the manuscript, it seems like that the manipulation of object in Chladni plate is achieved via linearly combined sequence notes which can produce basic Chladni figures. It is reasonable to use this method to manipulate a single object. However, for multiple objects, it is questionable that the same sequence notes can be used to independently and simultaneously control the movements. It is still unclear about why the independent and simultaneous control of multiple objects can be realized in this method. The movement of multiple objects in the same acoustic field can be different because of the different initial locations. Even though the experimental results in Fig. 1c and Fig. 2 seem like that the objects move independently, it is quite possibly a coincidence that the moving trajectories of the three particles just matched with the simple letters "S C I". In order to eliminate the concern of coincidence, it is suggested to demonstrate moving multiple objects independently and simultaneously along much more complicated traces, like the one shown in Fig. 1D.

We apologize that the text is not clear. A new Video S2 has been added to illustrate the control algorithm with two particles following different trajectories. The key point of our method is that we recalculate the weights of all notes after every step during the manipulation to generate the next note. We have feedback control (based on machine vision detected particle positions using the camera) that evaluates the positions of all the particles after every played note. The control algorithm then assigns weights to all notes, and determines the note to be played in the next step, where the previous assigned weight of each note will also be taken into account in the current decision. In this way, a note sequence is created that generally moves multiple particles to their corresponding targets, resulting in largely independent trajectories. The mathematical formulation is specified in details in the section “Control algorithm” in Methods.

- **We have also clarified the main text (lines 64-74):**

“We use the model to control the motion of the objects on the plate by iteratively generating a note sequence that, on average, moves the objects towards their desired directions. In each iteration, the computer finds a positive combination of all the modelled displacement fields (Extended Data Fig. 1) such that the net motion for all the objects at their current position is guided towards the desired direction. The positive combination is interpreted as a weight for each note. The computer then converts these weights into a sequence of notes, where the relative rate of occurrence of a note in the sequence is proportional to the weight of that note. For example, if in the first iteration notes A, B and C have the weights 3/6, 2/6 and 1/6,

respectively, then the implied note sequence to be played is ABAABCABAABC... corresponding to the weights. In practice, the note sequence is updated in every iteration based on the object locations and desired directions. The details of the control method are explained thoroughly in Methods, as well as in Movie S2.”

- To eliminate the concern of coincidence, we carried out additional experiments where two objects followed the A? trajectory, as shown in the new Video S10.

Also, for the results show in Fig. 2B, the final positions of the six objects can be matched by a circle or a rectangular, rather than a triangle it stated in the manuscript. To avoid misunderstanding, it is suggested to use more objects in the experiment for indicating the nodal lines more clearly.

We agree that the final object positions in the original Fig. 2B can be interpreted differently leading to unnecessary misunderstanding, even though the triangle was the target of the control algorithm. To avoid misunderstanding, we have aligned all 6 particles (RFID chips) in a straight line (see Extended Data Fig. 12). Such a line is not a natural nodal line of the plate (see Extended Data Fig. 1), but a target pattern for the control algorithm. For better demonstration, we also carried out pattern transformation experiment, as shown in the new Fig. 2B and Video S6.

3. The resolution of position control is not clear. From the result shown in Fig. 1 and Fig. 2, it looks like the error is quite large for objects in size of 600 μm .

In addition, as stated in the manuscript, the movement of objects can be coupled together when they become close to each other (<5 mm). Actually, comparing to the size of objects (600 μm), the distance threshold is quite large, such that the precision of the position control in this method is questionable. The resolution of position control in this method should be given and proved via experiments.

We have carried out experiments to characterize the manipulation accuracy (Extended Data Fig. 9). The accuracy for line tracking of a square shape is 4 μm and the standard deviation is 125 μm , where the pixel resolution of the camera in the object space (60 μm) has a major contribution to the error. The motion accuracy is also affected by the camera resolution. The smallest incremental motion detected by high speed camera and microscope is 1 pixel, corresponding to around 4 μm (Video S11).

We have also conducted experiments to check the influence of particle proximity at different initial centre-to-centre distances of 0.8 mm, 1.0 mm, 2.0 mm and 2.5 mm. For each initial distance, we placed two particles at 10 randomly generated positions on the plate, then the controller tries to separate them by increasing the distance to 10 mm. The results show that the separation is 100% successful with initial distance of 2.5 mm (Extended Data Fig. 11).

- We have added a paragraph in the Main Text (lines 144-154):

“We quantified the achievable control accuracy of the system by performing line tracking experiments and measuring the tracking error from the reference line. The error histogram with all 59 notes in Extended Data Fig. 9 shows close to a normal distribution. The measured data show an accuracy of 4 μm and a standard deviation of 125 μm . The standard deviation is equivalent to about 20 % of the size of the 600 μm solder ball used in the test. Each pixel in our top view camera corresponds to approximately 60 μm in the object space, which can already account for a significant part of the tracking error. We also quantified the smallest inducible particle motion using a side view high-speed camera, by progressively reducing signal duration. Video S11 shows an experiment where a single F8 note (5588Hz) with a duration of 2.5 ms was played. The detected motion is one pixel, equivalent to 4 μm in object

space. Therefore, the accuracy of our system could be further improved by increasing the top view resolution from the current 60 μm and reducing the duration of the notes.”

- We have also added a section “Resolution and accuracy characterization” in the Methods (lines 436-448).
- A paragraph has been added in main text (lines 156-159):
“We have measured the minimum distance that the controller can separate two adjacent objects (Extended Data Fig. 11). The controller was 100% successful at separating two objects when the initial centre-to-centre distance was 2.5 mm, but the success rate dropped to 30% when the two objects started at a distance of 0.8 mm.”

Moreover, manipulation of cells in lab-on-chip system and microfluidics requires much higher resolutions (1~10 μm). To support the claim 2 that the method can be useful in micro-scale manipulation, like cell culturing, lab-on-chips and microfluidics, it needs to demonstrate the ability of this method experimentally, at least to show that the method is able to manipulate objects in cell size independently and precisely.

The last part of the abstract is perspective according to the guideline of the original submission, and are not claims. We have moved the last two sentences of the abstract to discussion at the end of the main text. As shown in the Video S11, the position resolution of the system is 4 μm (1 pixel).

The current work focuses on the working principle to build a proof-of-concept apparatus. For potential cell or micro particle manipulation, we plan to build new apparatus for microfluidic systems, where the scale of the whole system will be smaller. In control applications, the controllable resolution is often related to sensor resolution. Currently, our optical resolution is about 60 μm in a large 50 \times 50 mm workspace, and we reached an accuracy of 4 μm and a stand deviation of 125 μm in line tracking. For cell manipulation, the operational environment will be much smaller, usually in a few millimetres, and the optical resolution will be one to few micrometres. So we expect our approach is relevant in manipulating smaller objects with micrometre accuracy, provided that the new apparatus is properly designed to allow out-of-nodal-line control. This new application requires substantial amount of work and a new research project.

References

1. Chen, P. *et al.* Microscale Assembly Directed by Liquid-Based Template. *Adv. Mater.* 1–6 (2014). doi:10.1002/adma.201402079
2. Tuan, P. H. *et al.* Exploring the distinction between experimental resonant modes and theoretical eigenmodes: From vibrating plates to laser cavities. *Phys. Rev. E - Stat. Nonlinear, Soft Matter Phys.* **89**, 1–9 (2014).
3. Tuan, P. H. *et al.* Exploring the resonant vibration of thin plates: Reconstruction of Chladni patterns and determination of resonant wave numbers. *J. Acoust. Soc. Am.* **137**, 2113–2123 (2015).
4. van Gerner, H. J., van der Weele, K., van der Hoef, M. A. & van der Meer, D. Air-induced inverse Chladni patterns. *J. Fluid Mech.* **689**, 203–220 (2011).

NCOMMS-16-07995A

REVIEWERS' COMMENTS:

Reviewer #1 (Remarks to the Author):

The questions raised in my review have been clearly answered. In particular, the application of the technique to a much wider range of objects demonstrates its generality. The addition of the new error analysis also makes it clear how this technique performs so it can be compared and contrasted with other manipulation strategies.

Reviewer #2 (Remarks to the Author):

This is a much improved manuscript with better explanations and more complete analysis. My remaining comments are relatively minor.

In the response to reviewers:

- The point about the slowness of manipulation is not a criticism so much as it is an opportunity to comment on tradeoffs or equivalences in the speed of manipulation as you reduce the number of actuators or increase the number of objects. One might opt for more actuators (e.g., 4 or 6 as in [24], [25]) to get faster manipulation. The time-sequencing of notes in this paper is akin to the time-sequencing of notes in [24] (see discussion below), and as the number of objects increases, manipulation generally gets slower.

- "nodes are understood as points where the amplitudes ... (are at a local minimum)." This should be clarified in the paper, since this is not a usual definition of a "node."

In the paper itself:

- In the first paragraph of the main text, it is mentioned that references [24] and [25] control the motion of a single particle. In fact, the authors of [24], in two different conference papers in 2001, show how to control the motions of multiple parts simultaneously using vision feedback, much like the present paper. In their work, if there are n parts to be controlled, they control each in sequence, at each step applying a field that is relatively localized to one part while having little effect on the others. Vision feedback is used to determine the direction of motion of each part. The resulting controller is similar to the control method in the present paper, in that the resulting sequenced fields are similar to a positive combination of fields.

Also, the authors of [25] control several parts simultaneously. Their approach is to design shaking plate motions that create desired vector fields with appropriate vectors at the locations of the individual parts. These can be used without vision feedback for tasks requiring negative divergence vector fields.

The paper should more accurately characterize the contributions of this prior work.

- Instead of saying that the note played is the one "most likely to move the objects towards the desired directions," it would be more accurate to say that the note played is the one "that is expected to most closely achieve the desired motions of the objects."

- The word "Remarkably" should be eliminated. "Remarkably" implies that something unexpected happened, whereas the whole method was designed to allow u_n to model the motion of the particles away from the "nodal" lines.

- The description of the "positive combination" is unnecessarily unclear. Also, the description (5) in the control algorithm section is difficult to read/interpret. In this respect, the new video S2 is helpful for understanding. It appears that a weighted combination is calculated, then the note with the highest weight is played. This should be stated clearly first. Only after that might it be appropriate to say that a time sequence achieving a weighted average is played. But even then, this can be confusing, since the weighted combination appears to be recalculated after each 0.5 s step.

On first reading of the main text, the reader is likely to conclude that a weighted combination is calculated, then a sequence of notes is played achieving time ratios equal to the weight ratios before the next weighted combination is calculated. This appears not to be true.

There is no reason for the description of the control algorithm not to be clearly worded.

- The "typical displacement" used to choose the amplitude should be clearly defined, even in the main text. It appears to be just the average displacement of the balls over all the balls' motions.

- The "resonant peaks" are not well defined nor are they used anywhere in the paper, making their appearance in the "For modelling of..." paragraph a non-sequitur, and confusing. The paper says "This confirms that there is a relationship between the resonances of the plate and the observed movement of the objects," but no such relationship is referred to in the paper. Figure 3B should either be made more relevant to the paper (I believe this will be harder to do, and I don't recommend it), or dropped or demoted to the extended data. If it is not dropped then "inverse amplitude" should be more carefully defined (amplitude of what?), and the relevance of a sound pressure level reading and the importance of resonances (and even the meaning of resonance in the context of a Chladni plate) should be made clear.

- It is claimed that "the vibration is less sensitive to the mass on the plate..." Less sensitive than what, and why would we care? It suffices to say that for the small masses tested on top of the plate (salt, solder balls, etc.), there is no noticeable difference in the plate's vibration patterns. For heavier objects (e.g., a full bottle of water), I doubt that would be the case.

- The discussion in lines 145-150 is unnecessarily confusing:

-- "accuracy" is undefined

-- The standard deviation is much higher than the "accuracy," again calling into question what "accuracy" means.

-- 4 μm accuracy is claimed, but two sentences later says that the top view camera has a pixel size of 60 μm , so it would be impossible to measure "accuracy" at this low level.

Some of this is (somewhat) clarified in the later discussion, but this paragraph can be more clearly worded.

- "Below the threshold distance...", line 165: It would be clearer to say that the controller changes its objective function in this case.

- line 170: Define "front edge" and "back edge" in the context of your planar side view of the object.

- line 192, concerning "accurate position predictions of the objects": Here it is implied that predicting for short periods of time means better predictions, but this is meaningless. If the period of time = 0, then there is zero error (just use the vision estimate of the current position). The issue is how much signal there is in the noise, and you cannot increase that by shortening the prediction time.

- The sentence in lines 196-7 is awkward. Change to "In this study, we have shown that the motion of multiple particles can be controlled independently using a Chladni plate with only one actuator," or something.

- line 432: 10,000 should have units fps.

- line 703 should refer to Fig 13A.

Reviewer #3 (Remarks to the Author):

The authors has appropriately addressed my comments. The article can be accepted as it is.

Response to Referees for NCOMMS-16-07995A

We thank the reviewer for the helpful suggestions and comments. In the following, we address the reviewer's comments point by point, where our responses are in bold.

Notice: the line number referred below is valid only with the "Simple Markup" or "No Markup" settings in "Track Changes" in Microsoft Word.

Reviewer #2:

This is a much improved manuscript with better explanations and more complete analysis. My remaining comments are relatively minor.

In the response to reviewers:

1. The point about the slowness of manipulation is not a criticism so much as it is an opportunity to comment on tradeoffs or equivalences in the speed of manipulation as you reduce the number of actuators or increase the number of objects. One might opt for more actuators (e.g., 4 or 6 as in [24], [25]) to get faster manipulation. The time-sequencing of notes in this paper is akin to the time-sequencing of notes in [24] (see discussion below), and as the number of objects increases, manipulation generally gets slower.

We thank the reviewer for the comment.

2. "nodes are understood as points where the amplitudes ... (are at a local minimum)." This should be clarified in the paper, since this is not a usual definition of a "node."

Thanks for the comments. We have clarified this in the main text (lines 163-165):

"In a centrally actuated Chladni plate, the nodal lines are points where the amplitude of a time-harmonic vibrating plate reaches its local minimum, occurring at all frequencies rather than at specific frequencies."

In the paper itself:

3. In the first paragraph of the main text, it is mentioned that references [24] and [25] control the motion of a single particle. In fact, the authors of [24], in two different conference papers in 2001, show how to control the motions of multiple parts simultaneously using vision feedback, much like the present paper. In their work, if there are n parts to be controlled, they control each in sequence, at each step applying a field that is relatively localized to one part while having little effect on the others. Vision feedback is used to determine the direction of motion of each part. The resulting controller is similar to the control method in the present paper, in that the resulting sequenced fields are similar to a positive combination of fields.

Also, the authors of [25] control several parts simultaneously. Their approach is to design shaking plate motions that create desired vector fields with appropriate vectors at the locations of the individual parts. These can be used without vision feedback for tasks requiring negative divergence vector fields.

The paper should more accurately characterize the contributions of this prior work.

We thank the reviewer for the detailed comments. We have replaced the ref. [24] with two publications of the authors in 2001. The work of ref. [24] and related work and [25] uses vector fields like us, but with significant differences. The motion mechanism of those works is friction based lateral vibrations, vs. our transverse vibrations. Ref. [24] and related work use a motion

primitive (local velocity field) to move multiple particles one by one in sequence. Ref [25] can control multiple particles simultaneously but not independently. Our work can control multiple particles both simultaneously and independently despite of having just one actuator, compared to four in ref. [24] (and related work) and six in ref [25].

We have cited those works in more details in the main text (lines 36-38) and replaced ref. [24] with the two 2001 conference papers of the authors:

“Four actuators have also created a local velocity field to control the motion of multiple individual objects one by one, and six actuators can create programmable velocity fields to move multiple objects in a dependent manner”

4. Instead of saying that the note played is the one "most likely to move the objects towards the desired directions," it would be more accurate to say that the note played is the one "that is expected to most closely achieve the desired motions of the objects."

Thanks for the comments, we have modified the main text accordingly (lines 58-59):

“finds the frequency of a note that is expected to most likely achieve the desired motion”.

5. The word "Remarkably" should be eliminated. "Remarkably" implies that something unexpected happened, whereas the whole method was designed to allow u_n to model the motion of the particles away from the "nodal" lines.

Thanks for the comments. We have replaced the word “Remarkably” with “In summary, we modelled...” in the main text line 75.

6. The description of the "positive combination" is unnecessarily unclear. Also, the description (5) in the control algorithm section is difficult to read/interpret. In this respect, the new video S2 is helpful for understanding. It appears that a weighted combination is calculated, then the note with the highest weight is played. This should be stated clearly first. Only after that might it be appropriate to say that a time sequence achieving a weighted average is played. But even then, this can be confusing, since the weighted combination appears to be recalculated after each 0.5 s step.

On first reading of the main text, the reader is likely to conclude that a weighted combination is calculated, then a sequence of notes is played achieving time ratios equal to the weight ratios before the next weighted combination is calculated. This appears not to be true.

There is no reason for the description of the control algorithm not to be clearly worded.

Thanks for the comments, we have clarified the description of the control algorithm by focusing the discussion on calculating the weights and updating the accumulated weights at each time step. We have modified the text in the main text (lines 77-86):

“We control the motion of the objects on the plate by repeatedly measuring the position of the objects and use the model to choose a note that moves the objects towards their desired directions. In each time step, the computer finds a linear combination of all the modelled displacement fields (Supplementary Fig. 1) such that all the weights are positive and the net motion for all the objects at their current position is guided towards the desired direction. The new weight of each note is then added to the previously accumulated weight of that note. Then the note with the highest accumulated weight is played and its accumulated weight is reset to zero. The procedure is repeated in the following time steps until the targets are reached. As a result, notes which have generally large weights will be played more often, but occasionally notes with small but non-zero weights will also be played.”

We have also updated the text in the Control algorithm section of Methods, by adding the concept of time step in notations (lines 431-460).

7. The "typical displacement" used to choose the amplitude should be clearly defined, even in the main text. It appears to be just the average displacement of the balls over all the balls' motions.

Thanks for the comments. We have replaced “typical displacement” with “75% quantile absolute displacement” in the main text line 122.

8. The "resonant peaks" are not well defined nor are they used anywhere in the paper, making their appearance in the "For modelling of..." paragraph a non-sequitur, and confusing. The paper says "This confirms that there is a relationship between the resonances of the plate and the observed movement of the objects," but no such relationship is referred to in the paper. Figure 3B should either be made more relevant to the paper (I believe this will be harder to do, and I don't recommend it), or dropped or demoted to the extended data. If it is not dropped then "inverse amplitude" should be more carefully defined (amplitude of what?), and the relevance of a sound pressure level reading and the importance of resonances (and even the meaning of resonance in the context of a Chladni plate) should be made clear.

Thanks for the comments, we have clarified the frequency properties of the plate and added more detailed discussion on the relation between frequencies and manipulation amplitudes.

We have updated the main text (lines 118-134):

“The frequency dependence is far more complicated, and is related to the resonances of the plate: at resonance, much smaller actuation amplitudes are needed to move the particles. For controlled manipulation, we want to keep the particle displacement relatively constant for all notes. To do this, we keep the duration of the notes constant (0.5 s), and algorithmically adjust the amplitude of each note such that the 75% quantile absolute displacement, taken over approximately 100 particles distributed on the plate, is close to 175 μm for all notes. We call this adjusted amplitude the nominal amplitude for note n (see section Nominal amplitude in Methods). Peaks in the inverse of the nominal amplitude (Figure 3b) are expected to correspond to the resonant peaks of the plate.

We use two alternative methods to find the resonances of the plate: acoustic characterizations and theoretical computations. Figure 3b compares the resonant peaks obtained using all three methods. The resonant peaks in inverse nominal amplitude and acoustic characterization agree well with each other, and the theory agrees with the two experimental methods with a similar accuracy as has been achieved before. This confirms that there is a relationship between the resonances of the plate and the observed movement of the objects on the plate, which provides a quick way to estimate the nominal amplitude for each note from acoustic characterizations or theory. On the other hand, it also shows that objects can be moved when the plate is not in resonance, if the plate can be actuated with sufficient power.”

9. It is claimed that "the vibration is less sensitive to the mass on the plate..." Less sensitive than what, and why would we care? It suffices to say that for the small masses tested on top of the plate (salt, solder balls, etc.), there is no noticeable difference in the plate's vibration patterns. For heavier objects (e.g., a full bottle of water), I doubt that would be the case.

Thanks for the comments, we removed the sentence.

10. The discussion in lines 145-150 is unnecessarily confusing: - "accuracy" is undefined - The standard deviation is much higher than the "accuracy," again calling into question what "accuracy" means.

4 μm accuracy is claimed, but two sentences later says that the top view camera has a pixel size of 60 μm , so it would be impossible to measure "accuracy" at this low level.

Some of this is (somewhat) clarified in the later discussion, but this paragraph can be more clearly worded.

Thanks for the comments. We have removed the “accuracy” and replaced it with “mean error” and added a sentence to clarify that it is below sensor resolution in the main text (lines 152-154): “The measured data show a mean error of 4 μm and a standard deviation of 125 μm . The mean error is below the resolution of our top view camera.”

11. "Below the threshold distance...", line 165: It would be clearer to say that the controller changes its objective function in this case.

We have clarified the main text (lines 199-201):

“Below the threshold distance, the distance penalty is added to the cost function of the controller, which tries to move the objects away from each other (see sections Control algorithm and Manipulation experiments in Methods).”

12. line 170: Define "front edge" and "back edge" in the context of your planar side view of the object.

Thanks for the comments. We have replaced the usage of “front edge” and “back edge” with “left edge” and “right edge” in the main text (lines 205-208).

13. line 192, concerning "accurate position predictions of the objects": Here it is implied that predicting for short periods of time means better predictions, but this is meaningless. If the period of time = 0, then there is zero error (just use the vision estimate of the current position). The issue is how much signal there is in the noise, and you cannot increase that by shortening the prediction time.

Thanks for the comments. We have deleted the sentence.

14. The sentence in lines 196-7 is awkward. Change to "In this study, we have shown that the motion of multiple particles can be controlled independently using a Chladni plate with only one actuator," or something.

Thanks for the comments. We have changed accordingly, which is currently the lines 231-232.

15. line 432: 10,000 should have units fps.

Thanks, we have corrected it.

16. line 703 should refer to Fig 13A.

Thanks, corrected.